# Lysates of *Methylococcus capsulatus* Bath induce a lean-like microbiota, intestinal FoxP3$^+$RORγt$^+$IL-17$^+$ Tregs and improve metabolism

Benjamin A. H. Jensen [1,2,3,14✉], Jacob B. Holm[1,12,13], Ida S. Larsen[1,2,13], Nicole von Burg[4,13], Stefanie Derer [5], Si B. Sonne[1], Simone I. Pærregaard [1,4], Mads V. Damgaard [1,6], Stine A. Indrelid[7], Aymeric Rivollier[4], Anne-Laure Agrinier[2], Karolina Sulek[6], Yke J. Arnoldussen[7], Even Fjære[8], André Marette [2], Inga L. Angell[7], Knut Rudi [7], Jonas T. Treebak [6], Lise Madsen[1,8], Caroline Piercey Åkesson [9], William Agace[4,10], Christian Sina[5], Charlotte R. Kleiveland[7], Karsten Kristiansen [1,11,14✉] & Tor E. Lea [7,14✉]

Interactions between host and gut microbial communities are modulated by diets and play pivotal roles in immunological homeostasis and health. We show that exchanging the protein source in a high fat, high sugar, westernized diet from casein to whole-cell lysates of the non-commensal bacterium *Methylococcus capsulatus* Bath is sufficient to reverse western diet-induced changes in the gut microbiota to a state resembling that of lean, low fat diet-fed mice, both under mild thermal stress (T22 °C) and at thermoneutrality (T30 °C). Concomitant with microbiota changes, mice fed the *Methylococcus*-based western diet exhibit improved glucose regulation, reduced body and liver fat, and diminished hepatic immune infiltration. Intake of the *Methylococcu*-based diet markedly boosts *Parabacteroides* abundances in a manner depending on adaptive immunity, and upregulates triple positive (Foxp3$^+$RORγt$^+$IL-17$^+$) regulatory T cells in the small and large intestine. Collectively, these data point to the potential for leveraging the use of McB lysates to improve immunometabolic homeostasis.

[1] Laboratory of Genomics and Molecular Biomedicine, Department of Biology, Faculty of Science, University of Copenhagen, Copenhagen, Denmark. [2] Department of Medicine, Faculty of Medicine, Cardiology Axis of the Québec Heart and Lung Institute, Laval University, Laval, QC, Canada. [3] Novo Nordisk Foundation Center for Basic Metabolic Research, Faculty of Health and Medical Sciences, University of Copenhagen, Copenhagen, Denmark. [4] Mucosal Immunology, Department of Health Technology, Technical University of Denmark, Copenhagen, Denmark. [5] Institute of Nutritional Medicine, University Hospital Schleswig-Holstein, Campus Lübeck, Lübeck, Germany. [6] Novo Nordisk Foundation Center for Basic Metabolic Research, Integrative Metabolism and Environmental Influences, Faculty of Health and Medical Sciences, University of Copenhagen, Copenhagen, Denmark. [7] Faculty of Chemistry, Biotechnology and Food Science, Norwegian University of Life Sciences, Oslo, Norway. [8] Institute of Marine Research, Bergen, Norway. [9] Department of Anatomy and Pathology, Faculty of Veterinary Medicine, Norwegian University of Life Sciences, Oslo, Norway. [10] Immunology Section, Department of Experimental Medical Science, Lund University, Lund, Sweden. [11] Institute of Metagenomics, BGI-Shenzhen, Shenzhen, P.R. China. [12] Present address: Clinical Microbiomics, Copenhagen, Denmark. [13] These authors contributed equally: Jacob B. Holm, Ida S. Larsen, Nicole von Burg. [14] These authors jointly supervised this work: Benjamin A. H. Jensen, Karsten Kristiansen, Tor E. Lea. ✉email: Benjamin.jensen@sund.ku.dk; kk@bio.ku.dk; tor.lea@nmbu.no

Gut microbes shape intestinal immunity[1] and increase the bioavailability of otherwise indigestible nutrients[2]. A well-balanced community structure is therefore essential for immunometabolic homeostasis, whereas aberrant gut microbiota compositions associate with numerous diseases, both within and outside the gastrointestinal tract[3].

While therapeutic implications of rebalancing a mistuned gut microbiota appear promising, inconsistent response rates in relation to both probiotics and fecal transfer studies, with occasional adverse events, emphasize the complexity of such approaches. One example relates to the otherwise promising probiotic candidate *Akkermansia muciniphila*[4], where negative effects have been seen in immunocompromised recipients[5,6]. Similarly, *Prevotella copri* aids in metabolizing fibers in healthy individuals[7] and protects against bacterial invasion in high fiber, chow-fed mice[8], yet associates with insulin resistance in pre-diabetic obese individuals and precipitates glucoregulatory impairments in diet-induced obese (DIO) mice[9].

An alternative to administering viable microbes is to utilize whole-cell lysates, or selected cell components, of nonliving bacteria. Apart from alleviating global energy demands, if used as a nutrient source, such components may also potently affect host physiology as recently reported for *A. muciniphila*[10] and *Bifidobacterium bifidum*[11]. In the latter example, cell surface polysaccharides of *B. bifidum* were used to induce peripheral immune-tolerance via generation of regulatory T cells ($T_{regs}$). The authors reported a pronounced increase in Foxp3+RORγt+ $T_{regs}$ ($_pT_{regs}$), specifically in lamina propria (LP) of the large intestine (LI)[11]. This cell type is believed to be induced by commensal microbes and has emerged as a potent $T_{reg}$ subset, exhibiting increased lineage stability and enhanced immunosuppressive capacity during intestinal inflammation compared to conventional Foxp3+RORγt- $T_{regs}$ ($_nT_{regs}$)[12].

RORγt is the canonical transcription factor controlling IL-17 expression; a pleiotropic cytokine with both proinflammatory and immune resolving actions depending on the eliciting cell type and physiological context[13]. Unfortunately, the studies describing $_pT_{reg}$ function did not measure IL-17 secretion. It therefore remains unknown whether these cells exhibit normal, reduced or increased IL-17 levels, and how this translates to host physiology. The impact of this predominantly colonic cell subset on host metabolism also remains unknown. Still, mounting evidence points towards the importance of intestinal IL-17 for controlling metabolic homeostasis[14,15]. IL-17+ $_pT_{regs}$ may therefore be leveraged as a 'dual hit' strategy to curb immunometabolic dysfunction and gastrointestinal disturbances based on the immune-regulatory capacity of $_pT_{regs}$ concomitant with the metabolic benefits of gut-delivered IL-17.

To this end, we hypothesized that environmental bacteria, who have not been under evolutionary scrutiny for host-microbe interactions, would provide an unexplored reservoir of immunomodulatory stimuli. In support of this hypothesis, the methanotropic noncommensal bacterium *Methylococcus capsulatus* Bath (McB) has previously been shown to interact with human dendritic cells (DC) modulating T cell responses in vitro[16], and to reduce inflammation and disease activity in dextran sulfate sodium (DSS)-induced mouse colitis[17]. However, the impact on host metabolism and intestinal immune cells as well as mucus dynamics were not addressed.

We accordingly explored the effect of using whole-cell lysates from McB as protein source to reshape immunometabolism and the aberrant gut microbiota of DIO mice. We show that McB lysates augment Foxp3+RORγt+IL-17+ triple-positive $_pT_{regs}$ in both SI- and LI-LP, and reset the obese microbiota concomitant with reversed key disease traits of diet-induced obesity.

## Results

### McB feeding reverses WD-induced gut microbiota changes and increases cecal SCFA levels.

To induce obesity and immunometabolic dysfunctions, C57BL/6JRj mice were initially fed an obesogenic WD. After 12 weeks of WD feeding, the mice were stratified into new groups based on weight, fat mass and glucoregulatory capacity (Supplementary Fig. 1a), and fed experimental WDs for an additional 6 weeks. While dietary fat is known to elicit reproducible and lipid-dependent alterations in the murine gut microbiome across a variety of different diet compositions[18], less is known about the microbiota-modulating impact of protein. Thus, to investigate if dietary protein (i.e., casein versus whole-cell bacterial lysates) would affect gut microbiota community structures, we analyzed freshly collected fecal samples before and throughout the dietary intervention. LFD and $WD_{REF}$ fed mice showed distinct gut microbiota profiles after 12 weeks of feeding (intervention baseline, week 12 + 0; Fig. 1A, B), including ~10-fold lower abundance of the health-promoting genera, *Parasutterella*[19] and *Parabacteroides*[20], countered by an equally increased abundance of the obesity-associated genus *Desulfivobrio*[21,22] (Fig. 1C) as well as a ~4-fold increase in the *Firmicutes* to *Bacteroidetes* (F/B) ratio (Fig. 1D). Interestingly, $WD_{CNTL}$ fed mice showed negligible changes in the microbiome signature during the 6 weeks of intervention (Fig. 1A–E; Supplementary Fig. 2a, b), suggesting that the added lipid source had limited influence on the intestinal ecology. In contrast to this observation, we noted a pronounced shift in bacterial composition in mice fed $WD_{McB}$. Within the first 2 weeks of treatment, the general community structure in these mice shifted towards that of their LFD-fed counterparts (Fig. 1A–E, Supplementary Table 3). We next asked if the observed taxonomical differences between groups related to alterations in the functional potential. SCFAs are main end products of metabolized fibers, and to a lesser extent amino acids escaping digestion in the SI, with vast impact on host physiology[23,24]. The highest levels of SCFAs are found in the cecum and proximal colon[25]. We therefore investigated if cecal SCFA levels were different between groups. We found a consistent increase in the levels of the three major as well as three minor classes of SCFAs in the cecum of $WD_{McB}$ fed mice compared to $WD_{CNTL}$ fed counterparts pointing towards not just taxonomically, but also functionally, discrete microbiota profiles in the two groups of mice, supporting a beneficial health impact of dietary inclusion of McB lysates (Fig. 1F, G).

### $WD_{McB}$ feeding stimulates induction of gut-specific regulatory T cells.

The intricate relationship between gut microbes and host immunity, combined with the immunoregulatory capacity of SCFAs[26], prompted us to investigate if the observed changes mediated by $WD_{McB}$ feeding were associated with immune alterations.

We accordingly analyzed the immune cell profile of SI-LP and LI-LP in a subset of experimental mice ($n = 6–10$/group) using multicolor flow cytometry focusing on phenotypic characterization of group 3 innate lymphoid cells (ILC3), natural killer (NK) cells and T cells (consult Supplementary Fig. 3a, b for gating strategies). Numbers of ILC3s, NK cells and T cell receptor (TCR)-γδ+ T cells, were similar between groups (Supplementary Fig. 3d–f). The same was true for the numbers of TCRβ+ CD4+ T cells, as well as the proportion of T helper ($T_H$)1-, $T_H$17-, and $_nT_{regs}$ cells (Fig. 2A, B; H, I). Interestingly, the proportion of $_pT_{regs}$ was more than 2- and 3-fold increased in LI- and SI-LP, respectively of $WD_{McB}$ fed mice compared to $WD_{CNTL}$ fed counterparts (Fig. 2C, J $p < 0.001$; S3G). Notably, this regulatory T cell subset has been shown to curb intestinal inflammation[12] and mediate immunological tolerance to the gut pathobiont

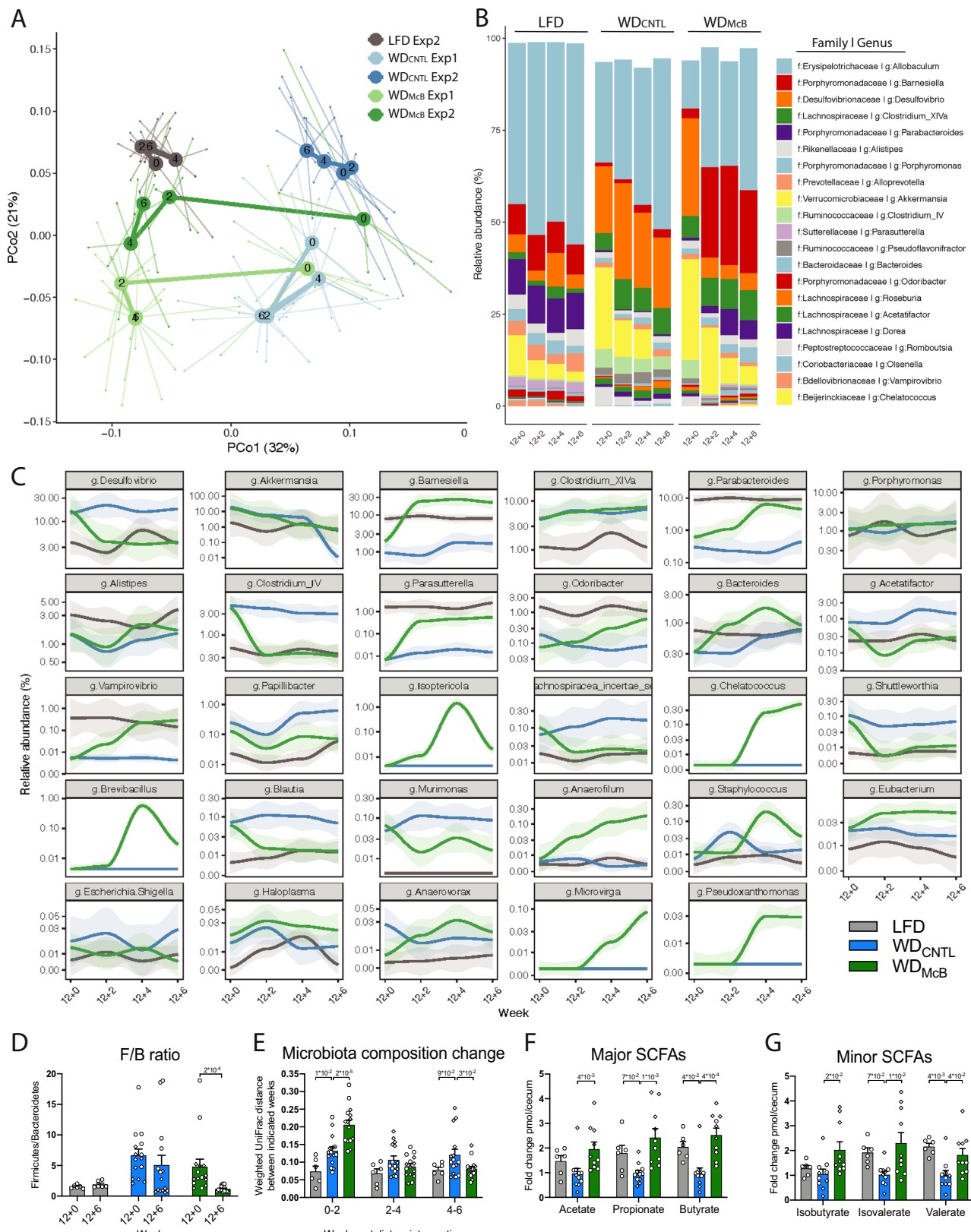

*Helicobacter hepaticus*, thereby protecting against $T_H17$-mediated barrier dysfunction and subsequent colitis[27]. Because RORγt is the hallmark transcription factor for $T_H17$ cell differentiation and essential for their IL-17 production, we next assessed if the $_pT_{regs}$ induced by the different diets were also capable of expressing IL-17. Indeed, ex vivo stimulated $_pT_{regs}$ produced substantial and

diet-dependent amounts of IL-17 protein, where $WD_{McB}$ feeding increased the proportion of IL-17+ cells within LI-LP $_pT_{regs}$ (Fig. 2D–G, $p < 0.001$; 2k-N; S3H). We confirmed gut-specificity of the $WD_{McB}$-induced $_pT_{regs}$ as only negligible amounts were observed in single-cell suspensions of liver homogenates obtained from 'weight-matched' mice fed the respective diets

**Fig. 1 McB feeding reverses WD-induced gut microbiota changes and increases cecal SCFA levels. A** Principle coordinate analysis (PCoA) using Weighted UniFrac distances of fecal microbiota sampled from first and second experiment biweekly during the dietary intervention period, as indicated by numbers post dietary intervention in centroids. The $WD_{CNTL}$ and $WD_{McB}$ groups were similar in microbiota composition prior to dietary intervention week $12 + 0$ (PERMANOVA $p = 0.88$ and $0.43$ in Exp1 and Exp2, respectively). At the end of each experiment, the microbiota composition was significantly different between these groups (PERMANOVA $p = 0.001$ and $0.002$ in Exp1 and Exp2, respectively). **B** Taxasummary of most abundant bacterial genera showing mean relative abundance in % of indicated family and genera in each group at indicated time points. **C** Deseq analysis of fecal bacterial genera abundances significantly regulated by McB intervention compared to the $WD_{CNTL}$ (p.adj. $< 0.05$). Relative abundance in % in each group and variation are shown for each regulated genus at the sampled time points. Fold-change and adjusted $p$ values of individual genera are indicated in Supplementary Table 3. **D** Relative Firmicutes/Bacteroidetes ratio of fecal samples of individual mice before ($12 + 0$) and after 6 weeks of dietary intervention ($12 + 6$). Statistical significance is indicated by $p$ value on Wilcoxon matched-pairs signed rank test. **E** Weighted UniFrac distance (instability test) between paired samples from indicated 2-weeks' interval post dietary intervention. Statistical significance is indicated by p value on RM two-way ANOVA with multiple comparison and Bonferroni's post-hoc. **D**, **E** LFD $n = 6$, $WD_{CNTL}$ and $WD_{McB}$ $n = 15$ per group. **F** Major short-chained fatty acids (SCFAs) in cecal content as fold-change pmol per cecum. Propionate data was tested by one-way ANOVA and Dunnett's multiple comparisons test. Acetate and butyrate data was tested by Kruskal–Wallis test and Dunn's multiple comparisons test. LFD $n = 6$, $WD_{CNTL}$ $n = 11$, and $WD_{McB}$ $n = 10$. **G** Minor SCFAs in cecal content as fold-change pmol per cecum. Isobutyrate and isovalerate data were tested by Kruskal–Wallis test and Dunn's multiple comparisons test. Valerate data were tested by one-way ANOVA and Dunnett's multiple comparisons test. LFD $n = 6$. $WD_{CNTL}$ and $WD_{McB}$ $n = 10$ per group. **D–G** Bars indicate group mean ± SEM and individuals data points in Exp1 (squares) and Exp2 (circles) with $p$-values $< 1 \times 10^{-1}$ between $WD_{CNTL}$ and indicated group using the specified statistical test.

prophylactically for 7 weeks (Supplementary Figs. 1c, 4a). $_pT_{regs}$ constituted $<1\%$ of all $CD4^+$ T cells in the liver, hence contrasting the $\sim3$ and 20% in SI- and LI-LP, respectively, of LFD- and $WD_{REF}$-fed mice and stunning $\sim12$ and 30% in similar sites of $WD_{McB}$-fed mice (Supplementary Fig. 4b). Mechanistically, LI-, but not SI-, LP-derived McB-induced $_pT_{regs}$ exhibited enhanced secretory capacity of the hallmark suppressive cytokine, IL-10, upon ex vivo stimulation (Fig. 2O, consult Supplementary Fig. 3c for gating strategy). Augmented IL-10 secretion was identified in both $_nT_{regs}$ and $_pT_{regs}$ populations in LI-LP of $WD_{McB}$-fed mice (Fig. 2O, P, $p < 0.05$ and $<0.001$, respectively), pointing towards enhanced immune regulation following $WD_{McB}$ feeding. Although the absolute number of IL-$10^+{}_{n/p}T_{regs}$ was similarly increased in SI-LP of $WD_{McB}$-fed mice (Supplementary Fig. 4c, d), the relative proportion of IL-$10^+$ cells within these $T_{regs}$ remained similar between groups (Fig. 2O, P). Collectively, these data corroborate that the phenotypic shift of enhanced secretory capacity was restricted to the colon of $WD_{McB}$-fed mice. The amount of $Ki67^+$ cells followed the patterns of $_pT_{reg}$ abundances (Fig. 2Q, R; Supplementary Fig. 4e, f, consult Supplementary Fig. 3c for gating strategy).

**$WD_{McB}$ mitigates diet-induced obesity.** The altered immune profile combined with a shift of the gut microbiota towards a state similar to that observed in lean LFD-fed mice, could potentially elicit crosstalk to glucoregulatory organs. To examine if McB lysates could reverse impaired glucose regulation, we performed OGTT and assessed GSIS concomitant with body mass composition in obese mice fed $WD_{REF}$ for 11 weeks and after 5 weeks of dietary intervention allowing for temporal analyses (Supplementary Fig. 1a). All mice were stratified into experimental groups based on their pre-intervention glucoregulatory capacity (Supplementary Fig. 4g, h). While the response to glucose challenge remained largely unaffected from week 11 to week $12 + 5$, regardless of experimental diets (Fig. 3A–C), both 5 h fasted insulin levels and glucose-stimulated insulin responses were significantly increased in mice fed $WD_{CNTL}$ (Fig. 3E, $p < 0.01$ & $p < 0.001$, respectively) in accordance with our previous report on time-dependent alterations in glucose regulation[28]. LFD- and $WD_{McB}$-fed mice were fully protected against this detrimental trajectory (Fig. 3D–F; Supplementary Fig. 4i, $p = 0.24$ and $0.68$, respectively), and $WD_{McB}$-fed mice further exhibited modestly improved insulin sensitivity based on 5 h fasted glycemia (Fig. 3C, $p < 0.05$) and intraperitoneal insulin tolerance test (Fig. 3G, $p < 0.05$).

Overall, weight development mimicked the glucoregulatory capacity. As such, $WD_{McB}$-fed mice exhibited stability of weight, fat mass and lean mass when changed to experimental diets, contrasting the continuous weight and fat mass development of $WD_{CNTL}$-fed mice (Fig. 3H, I and Supplementary Fig. 4j). The absence of weight gain was not explained by decreased feed intake, but rather appeared to be associated with enhanced fecal energy secretion (Fig. 3J, K).

Since obesity and impaired glucose regulation are tightly associated with NAFLD[29], we next subjected paraffin-embedded liver sections to histological evaluation. These analyses revealed both diminished steatosis and hepatocellular ballooning in $WD_{McB}$-fed mice compared to $WD_{CNTL}$-fed counterparts, where especially hepatocellular ballooning was arrested in (or returned to) a state reminiscent that of lean LFD-fed mice (Fig. 3L, M, $p < 0.05$). Importantly, hepatocellular ballooning is instrumental in the development of the more severe liver disease, NASH[29].

**$WD_{McB}$ feeding resets the hepatic lipidome and decreases hepatic immune infiltration alleviating NAFLD.** Based on the decreased NAFLD in $WD_{McB}$-fed mice housed at $T_{22°C}$ we designed a new experiment (study outline, Supplementary Fig. 1b) using a recently described[30] method where thermoneutral housing ($T_{30°C}$) potentiates NAFLD in WT C57BL/6 J mice fed an obesogenic diet for 20–24 weeks. To more thoroughly investigate the effect of $WD_{McB}$-feeding, we also redesigned the diets and omitted macadamia oil in the $WD_{CNTL}$ group, as this might lead to progression of obesity (Fig. 3) and related disorders. This new diet design entailed an increased fat/protein ratio in $WD_{McB}$ compared to $WD_{REF}$, due to phospholipids inherently present in bacterial lysates[31] (Supplementary Table 1). Despite the lower protein content in $WD_{McB}$ compared to both other diets, the relative amounts of indispensable amino acids were similar between groups (Supplementary Fig. 1f) and protein availability was well beyond critical levels, corroborated by similar lean mass to $WD_{REF}$-fed mice post diet intervention (Supplementary Fig. 5a). Still, $WD_{McB}$-fed mice exhibited significantly improved 5 h fasting insulin levels and decreased fat mass (Fig. 4A, B), despite weight maintenance and significantly increased energy intake compared to both LFD and $WD_{REF}$-fed mice (Supplementary Fig. 5b–d). The decreased body fat mass was accompanied by a diminished NAS, supported by both pathological evaluation of H&E stained liver sections (Fig. 4C, D) and hepatocytic lipid content assessed by Oil-Red-O staining (Fig. 4E–G). We additionally observed augmented adiponectin secretion (Fig. 4H),

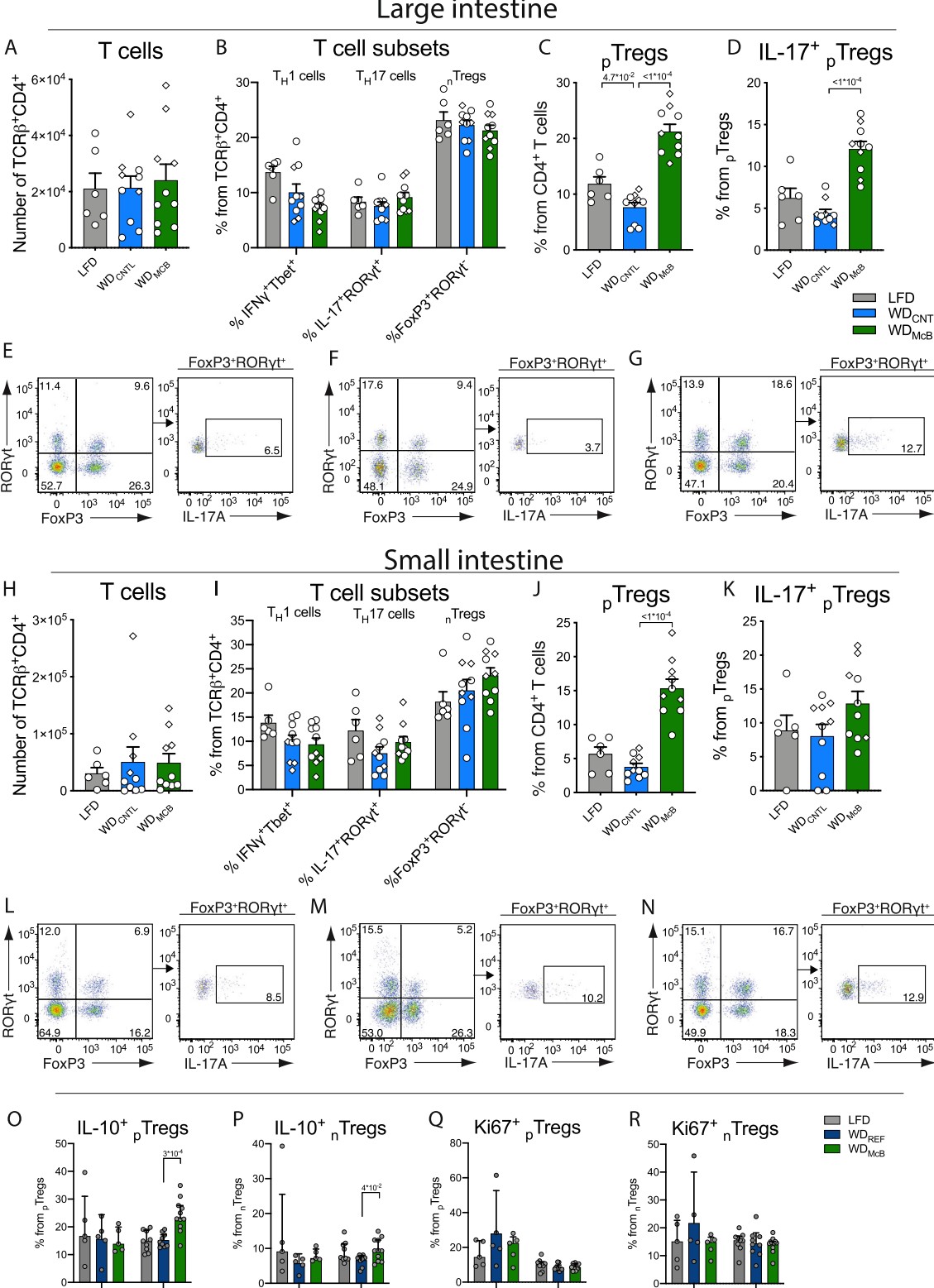

**Fig. 2 WD$_{McB}$ feeding stimulates induction of gut-specific regulatory T cells. A–D** Number of indicated cells in colon. **E–G** Representative plots of colonic TCRβ$^+$CD4$^+$ FoxP3$^+$ RORγt$^+$ $_p$T$_{regs}$ (left) and IL-17$^+$ $_p$T$_{regs}$ (right) in LFD (**E**), WD$_{CNTL}$ (**F**), and WD$_{McB}$ (**G**) group. **H–K** Number of indicated cells in small intestine. **L–N** TCRβ$^+$CD4$^+$ FoxP3$^+$ RORγt$^+$ $_p$T$_{regs}$ (left) and IL-17$^+$ $_p$T$_{regs}$ (right) in LFD (**L**), WD$_{CNTL}$ (**M**), and WD$_{McB}$ (**N**) group. **O–R** Percentage of indicated cells in SI- and LI-LP from 'weight-matched' mice housed at thermoneutrality. **A–N** Dots indicate individuals data points in Exp1 (squares) and Exp2 (circles) with n = 6 (LFD) or 10 (WD$_{McB}$ and WD$_{CNTL}$). **O–R** SI-LP $n = 5$ per group and LI-LP $n = 9$ (LFD and WD$_{REF}$ groups) or 11 (WD$_{McB}$). **A–D**, **H–K**, **O–R** Bars indicate group mean ± SEM and dots indicate individual data points. All $p$-values $<1 \times 10^{-1}$ between WD$_{CNTL}$ and indicated group by one-way ANOVA with multiple comparisons and Dunnett post-hoc are depicted.

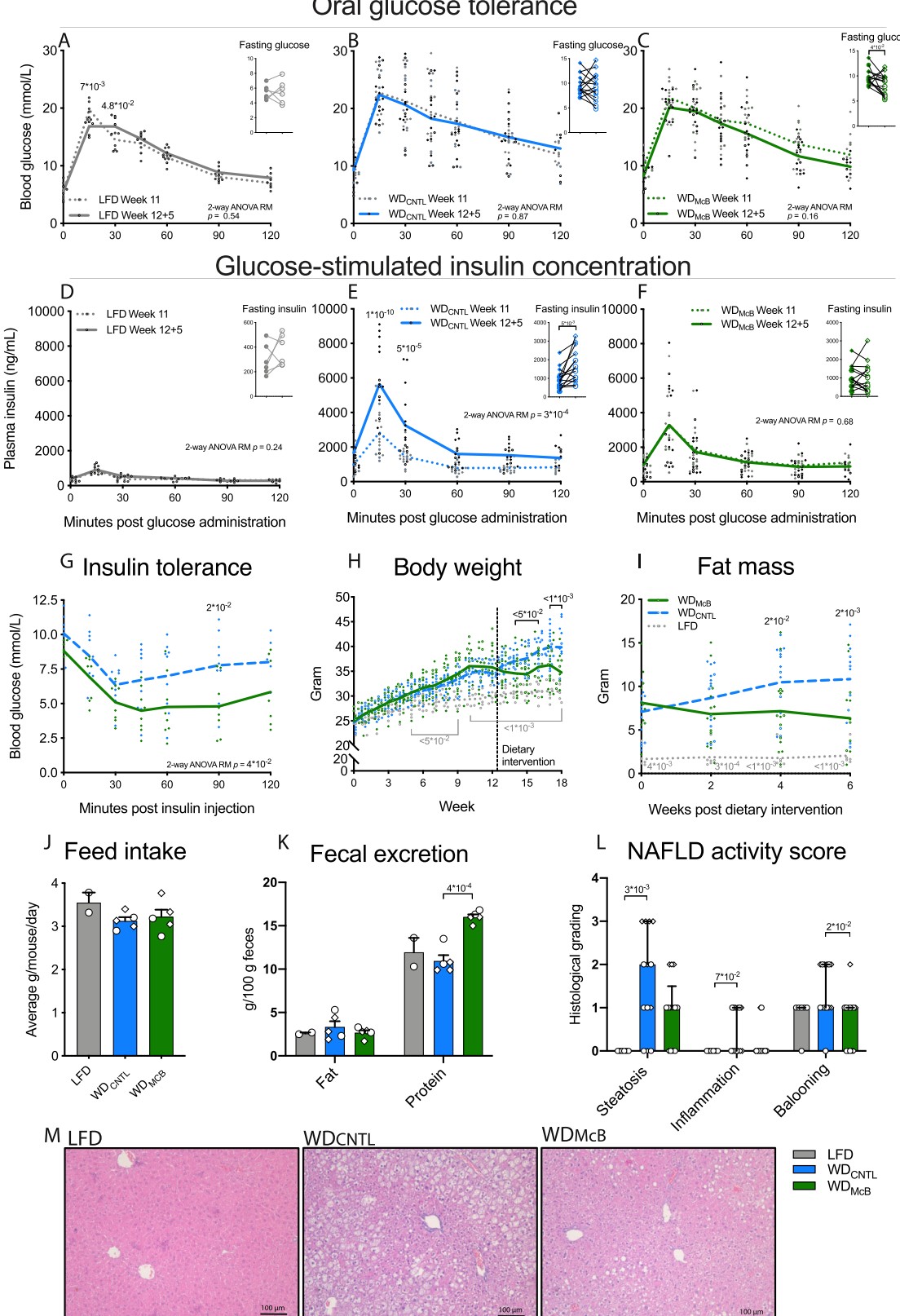

pointing towards improved insulin sensitivity in the WD$_{McB}$ group, further supported by the assessment of insulin tolerance and hepatic gene transcription activity of key metabolic enzymes (Fig. 4I, J). Of interest, we observed a >10-fold down-regulation of *Scd1* in the liver of WD$_{McB}$-fed mice, the hepatic expression of which is (a) regulated by the microbiota[32] and (b)

instrumental in de novo lipogenesis at the onset of metabolic syndrome[33].

We next assessed the hepatic lipidome by tandem mass spectrometry to elucidate if diminished NAFLD was associated with an altered lipid profile. Through comparison of WD$_{McB}$ and WD$_{REF}$ we identified 57 and 279 differentially regulated peaks in

**Fig. 3 Dietary intervention with McB blunts progression of insulin resistance and fat mass accumulation. A–C** Oral glucose tolerance test (OGTT) and 5 h fasting blood glucose prior to dietary intervention (Week 11) and 5 weeks post intervention (Week 12 + 5) of LFD (**A**), WD$_{CNTL}$ (**B**), and WD$_{McB}$ (**C**) groups. **D–F** Glucose-stimulated insulin concentration during OGTT and 5 h fasting insulin levels prior to dietary intervention (Week 11) and 5 weeks post intervention (Week 12 + 5) of LFD (**D**), WD$_{CNTL}$ (**E**), and WD$_{McB}$ (**F**) groups. **G** Intraperitoneal insulin tolerance test (ITT) 3 weeks post dietary intervention. **H** Body weight development. Mice were fed either LFD or WD$_{REF}$ the first 12 weeks followed by 6 week dietary intervention period. Dotted vertical line depicts intervention start. **I** Fat mass in gram through the dietary intervention period (Week 12 + 0 to 12 + 6) measured by MR scan. **J** Feed intake per cage as average grams per mouse during 48 h. **K** Fecal content of fat and protein. **L** NAFLD activity score based on hepatic steatosis grade (0–3), inflammation (0–2), and hepatocellular ballooning (0–3) graded by blinded histological assessment of liver tissue. **M** One representative H&E stained picture per group out of 6 (LFD) or 13 (WD$_{McB}$ and WD$_{CNTL}$) liver tissue sections. **A, F** $n = 6$ (LFD) and 15 (WD$_{McB}$ and WD$_{CNTL}$) except for F timepoint 60–120 min at Week 11 where $n = 14$, 12, and 12, respectively due to insufficient sample material. Statistical significance within each timepoint is indicated by p values at paired two-way ANOVA-RM with Bonferroni post-hoc test. Fasting glucose and insulin levels were evaluated by paired $t$-tests. Lines indicate group means and dots represents individual data points. **G** $n = 8$ (WD$_{McB}$) and 9 (WD$_{CNTL}$). WD$_{McB}$ was compared to WD$_{CNTL}$ by two-way ANOVA-RM with Bonferroni post-hoc test. Lines indicate group means and dots represents individual data points. **H–I** $n = 6$ (LFD) or 15 (WD$_{McB}$ and WD$_{CNTL}$). LFD and WD$_{McB}$ were compared to WD$_{CNTL}$ by two-way ANOVA-RM, adjusted for multiple comparisons by Dunnett's post-hoc. Lines indicate group means and dots represents individual data points. **J, K** Each data point represents the average of one cage. $n = 2$ (LFD) to 5 (WD$_{McB}$ and WD$_{CNTL}$). LFD and WD$_{McB}$ were compared to WD$_{CNTL}$ by one-way ANOVA, adjusted for multiple comparisons by Dunnett's post-hoc. Dot shapes indicate individuals data points in Exp1 (squares) and Exp2 (circles). **L** Bars represent median and interquartile range and dots represents individual data points. All $p$-values $< 1 \times 10^{-1}$ between WD$_{CNTL}$ and indicated group are depicted. Dot shapes indicate individuals data points in Exp1 (squares) and Exp2 (circles). LFD and WD$_{McB}$ were compared to WD$_{CNTL}$ by Kruskal–Wallis, adjusted for multiple comparisons by Dunn's post-hoc. $n = 6$ (LFD) or 13 (WD$_{McB}$ and WD$_{CNTL}$). **A–L** All $p$-values $< 1 \times 10^{-1}$ between WD$_{CNTL}$ and indicated group are depicted.

negative and positive ionization mode, respectively (Fig. 5A, B; Supplementary Fig. 5e, f, all FDR < 0.05). Of these, most classified lipids were changed with WD$_{McB}$ in the direction of LFD-fed mice (Fig. 5C, D). Notably, 57% of upregulated species were odd-chain fatty acids, whereas 80% of downregulated species represented lipids with even carbon numbers (Fig. 5C, D and Supplementary Table 4), hence supporting previous reports where odd- rather than even-chain fatty acids are inversely associated with human insulin resistance[9] and type 2 diabetes[34].

To estimate the functional consequences of an altered lipid profile, we used the Lipidmaps database to identify affected pathways and plotted the observed changes on a log$_2$ scale comparing both LFD and WD$_{McB}$ to WD$_{REF}$. The majority of affected pathways was similarly regulated in both direction and magnitude in LFD and WD$_{McB}$ mice compared to WD$_{REF}$ mice (Fig. 5E; Supplementary Fig. 5g). Notably, bile acids and ceramides, both of which were significantly downregulated in WD$_{McB}$-fed mice compared to WD$_{REF}$-fed counterparts, have been shown to mediate steatohepatitis by upregulation of IL-6 and TNF-α, respectively[35,36]. We therefore measured these hepatic cytokines and observed similarly reduced levels in both LFD and WD$_{McB}$ compared to WD$_{REF}$ (Fig. 5F).

A key feature of diet-induced liver pathologies, including NAFLD, is recruitment of newly activated immune cells capable of eliciting a proinflammatory immune response. This process is generally hampered in mice housed at mild thermal stress, which therefore fail to phenocopy human pathophysiology. However, thermoneutral housing recapitulates some human disease traits[37], which combined with HFD-feeding accentuates intrahepatic infiltration of proinflammatory Ly6$^{high}$ monocytes[38]. These monocytes interact with tissue resident T cells and play a central role in the pathogenesis of liver injury, hence representing an attractive therapeutic target to mitigate NAFLD development and to curtail associated pathologies[36].

We therefore subjected liver tissues from representative mice to immunological evaluation by immunohistochemistry and observed a marked decrease in both CD3$^+$ T cells and Ly6G$^+$ neutrophils in WD$_{McB}$-fed mice compared to their WD$_{REF}$-fed counterparts (Fig. 5G). Diminished hepatic immune infiltration was mirrored by increased levels of circulating IL-22, IL-18, and IL-17 in WD$_{McB}$-fed mice compared to their WD$_{REF}$-fed counterparts (Fig. 5H, $p < 0.01$, <0.05, and <0.05, respectively).

We next initiated a short term experiment in 'weight-matched' mice to evaluate the hepatic immune profile before obesity onset (Supplementary Figs. 1C; 5h, i). Surprisingly, this experiment revealed increased numbers of tissue resident Tim4$^+$ macrophages (i.e., Kupffer cells) in WD$_{McB}$-fed mice, suggesting that McB feeding either recruits or stimulate in situ proliferation of this key cell subset driving hepatic homeostasis (Fig. 5I). Kupffer cells are central to innate immunity and responsible for containment and clearance of foreign particles. Inflammatory activation of hepatic Kupffer cells potentiates obesity-associated insulin resistance, in part by recruiting neutrophils and T cells[39]. Yet, Kupffer cells exhibit tremendous plasticity in their activation program, with anti-inflammatory properties in their alternative activation state ameliorating hepatic steatosis[40]. While our staining panel did not allow us to identify the activation state of the enhanced Kupffer cell proportions, it is pertinent to note that none of the classically recruited cell types were altered in numbers (Supplementary Fig. 5j, k). Instead, we observed increased proportion of Ly6C$^+$ monocytes. The mean fluorescence intensity (MFI) within these monocytes was marginally lower in WD$_{McB}$-fed mice than in their WD$_{REF}$-fed counterparts (Fig. 5J). Newly recruited monocytes express high levels of Ly6C in their inflammatory state; an expression that is gradually downregulated in immune resolving alternatively activated cells[36,41,42]. Despite weight maintenance, WD$_{REF}$-fed mice exhibited ~50% increase in hepatic IL-17$^+$ γδ T cells (Fig. 5K), an immunological precursor for subsequent NAFLD controlled by the gut microbiota[43]. WD$_{McB}$-fed mice were fully protected from this trait, pointing towards extraintestinal regulation of innate immunity key to metabolic homeostasis.

**WD$_{McB}$ feeding reverses prolonged gut microbial dysbiosis and markedly improves colonic mucus production.** The improved hepatic phenotype prompted us to further investigate potential traits in the gut-liver axis. We initially assessed if the observed cytokine responses associated with improved gut health in mice housed at T$_{30°C}$, where inflammation is expected to be increased[37]. Indeed, WD$_{McB}$-fed mice were resistant to WD-induced colonic shortening closely associated with colonic inflammation (Fig. 6A).

To gain additional insights to the immunomodulating properties of McB lysates, we focused on mucus production and -function. Because mucin production is a constitutive process

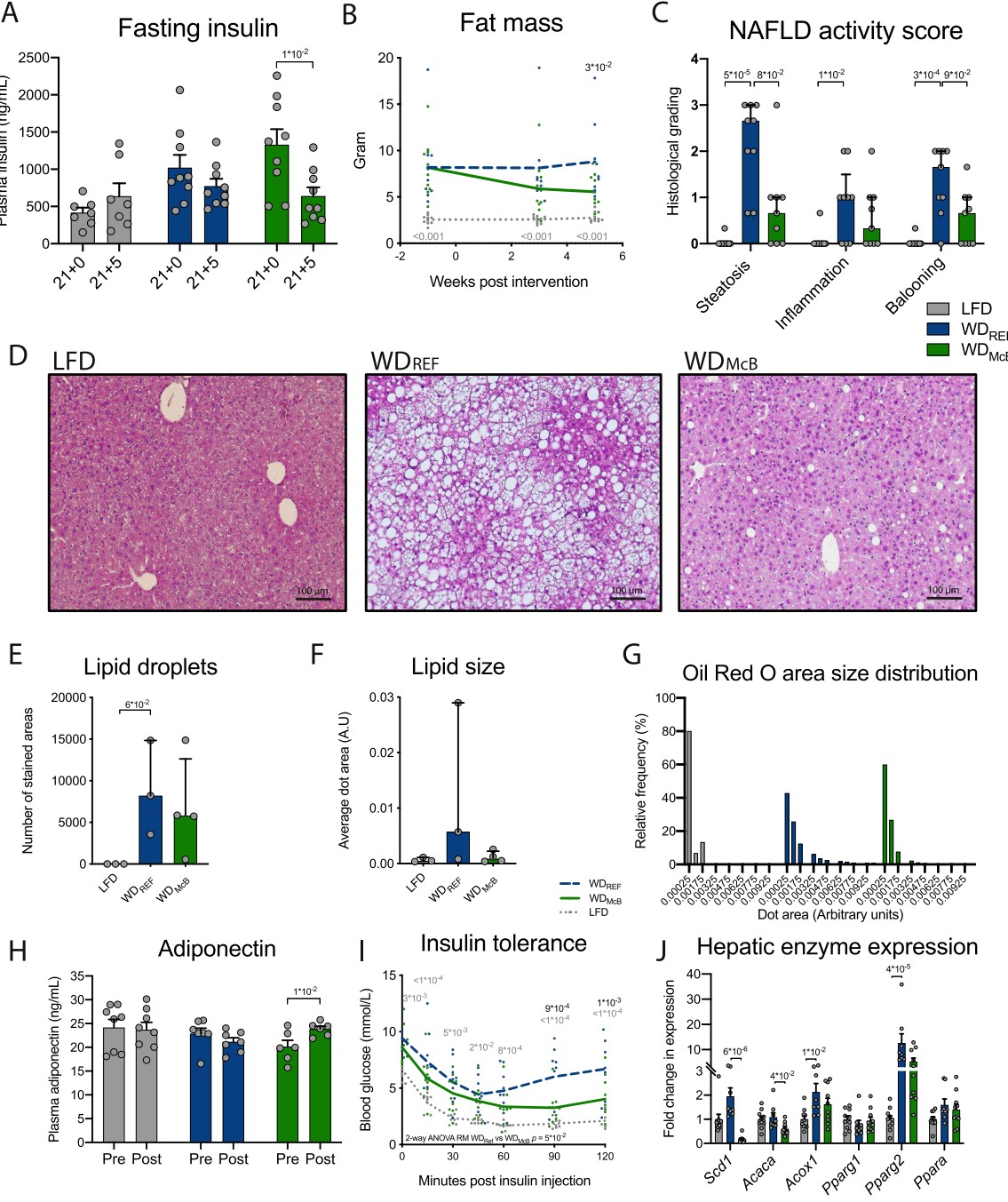

where both secretion and adherence are constantly ongoing and rapidly adjust to environmental changes, immunohistochemical labelling for various MUC epitopes may not fully recapitulate the physical properties of the mucus. We therefore applied specialized mucin histochemistry staining allowing us to differentiate between neutral and acidic mucins according to the net charge of each molecule. Acidic mucins were further separated into sulfomucins and sialomucins. Sections revealed that neutral mucins in crypt-residing goblet cells were consistently downregulated in WD$_{REF}$-fed mice compared to LFD-fed counterparts in three well defined segments of the colon; i.e. proximal, middle and distal area (Fig. 6B, D). WD$_{McB}$-feeding not only reversed this pattern in all three segments but even also enhanced the production of neutral mucins exceeding the levels found in LFD- fed counterparts. This is a remarkable finding considering the continuous intake of a westernized diet high in fat

and sucrose, known to hamper goblet cell function. We next evaluated crypt depth (CD) in stained sections. While the CD of the proximal and distal part of the colon was largely unaffected by diet, we observed increased CD in the middle segment of WD$_{McB}$- fed mice (Fig. 6C). WD$_{McB}$-feeding further enhanced the glycosylation pattern, particularly in the middle segment of colon, where this group exhibited a 3-fold increase in sulfomucins balanced by a similar (~2-fold) decrease in sialomucins compared to WD$_{REF}$-fed mice (Fig. 6E–G). No differences were observed between WD$_{REF}$ and LFD-fed mice, indicating that the reciprocal regulation of mucin glycosylation status was specific to WD$_{McB}$- feeding.

With focus on the dynamic interactions between gut immunity, mucin glycosylation and commensal microbes[44], we next assessed the gut microbiota composition in temporally separated samples. This was done to determine if the changes towards a gut

**Fig. 4 Dietary McB intervention improves diet-induced metabolic and hepatic phenotype after prolonged WD feeding. A** 5 h fasting plasma insulin levels before (Week 20) and after (Week 21 + 5) dietary intervention. Dots represents individual data point and bars represents mean ± SEM. $n = 7$ (LFD) or 9 (WD$_{REF}$ and WD$_{McB}$). LFD and WD$_{McB}$ were compared to WD$_{CNTL}$ by paired $t$-test. **B** Fat mass in grams measured by MR scan at indicated time points from dietary intervention. Dots represents individual data point and lines depicts mean. $n = 10$ (LFD and WD$_{McB}$) or 9 (WD$_{REF}$) per group. **C** NAFLD activity score separated in hepatic steatosis grade (0–3), inflammation (0–2) and hepatocellular ballooning (0–3) graded by blinded histological assessment of H&E stained liver tissue. Dots represents individual data point and bars represent median and interquartile range. $n = 10$ (LFD) or 9 (WD$_{REF}$ and WD$_{McB}$) per group. **D** One representative H&E stained picture per group out of 10 (LFD) or 9 (WD$_{REF}$ and WD$_{McB}$) liver tissue sections. **E** Number of lipid droplets in each liver sample per experimental group quantified by Oil Red O staining. Dots represents individual data point of 3 (LFD and WD$_{REF}$) and 4 (WD$_{McB}$) randomly chosen samples; bars represent median and interquartile range. **F** Average size of lipid droplets in each of 3-4 liver samples per experimental group quantified by Oil Red O staining in (E). Dots represents individual data point of 3 (LFD and WD$_{REF}$) and 4 (WD$_{McB}$) randomly chosen samples; bars represent median and interquartile range. **G** Distribution of lipid droplet size in % of all lipid droplets within each experimental group from **E**. **H** Plasma adiponectin concentration in 5 h fasted mice before (Week 20) and after (Week 21 + 5) dietary intervention. Dots represents individual data point and bars represents mean ± SEM. $n = 8$ (LFD), 7 (WD$_{REF}$), or 6 (WD$_{McB}$). **I** ITT after four weeks of dietary intervention (Week 21 + 4). Dots represents individual data point and lines depicts mean. $n = 7$ (LFD), 9 (WD$_{REF}$), or 10 (WD$_{McB}$). **J** Expression of key metabolic enzymes in liver tissue after 5 weeks of dietary intervention (Week 21 + 5) by RT-qPCR. Dots represents individual data point and bars represents mean ± SEM. $n = 10$ (LFD and WD$_{McB}$) or 8 (WD$_{REF}$) per group. **A, H** Statistical significance compared by paired $t$-test. **B, I** Statistical significance compared to WD$_{REF}$ by two-way ANOVA-RM, adjusted for multiple comparisons by Dunnett post-hoc. **C, E, F, J** Statistical significance compared to WD$_{REF}$ by Kruskal–Wallis test, adjusted for multiple comparisons by Dunn's post-hoc. **A–J** All p-values $< 1 \times 10^{-1}$ between WD$_{REF}$ and indicated group are depicted.

microbiota resembling that of lean LFD-fed mice was recapitulated in this intensified setup. In contrast to the first set of experiments, where we used cohoused mice shown to exhibit resilient microbiota profiles, we now employed single-housed mice to explore if the WD$_{McB}$-mediated community structures were persistent enough to induce consistent changes in the more dynamic communities of single-housed mice. Similar to our first experiments at T$_{22°C}$, we observed a normalization of the gut microbiota of WD$_{McB}$-fed mice, despite prolonged WD feeding prior to intervention (Fig. 6H–K). WD$_{McB}$-induced changes were surprisingly consistent with the first set of experiments, including a significantly lower F/B ratio (Fig. 6J), and a substantial reduction of *Desulfovibrio* abundance, countered by a similar bloom of the *Parasutterella* and *Parabacteroides* genera (Fig. 6K, Supplementary Table 5). Of note, the age-related increases in *Desulfovibrio* abundances recently reported[21] was confirmed in this study where the general magnitude in both WD$_{REF}$- and LFD-fed mice increased 2-3-fold over the 5 week intervention (Fig. 6K). WD$_{McB}$-feeding fully prevented this trajectory and paired analyses even revealed a diminished relative abundance of *Desulfovibrio* in these mice.

**McB lysates rely on adaptive immunity to favor *Parabacteroides* blooms.** To assess if WD$_{McB}$-induced microbiota alterations were a result of diminished obesity or altered immunity we next evaluated the impact of WD$_{McB}$-feeding in 'weight-matched' C57BL6/N mice fed the respective diets for 7 weeks (Supplementary Fig. 1c). In agreement with previous reports, WD$_{REF}$ feeding induced a ~3-fold decrease in *Parabacteroides* abundance independent of obesity, a feature that has been reported in several immune-competent mouse strains[45]. WD$_{McB}$ feeding not only prevented this trajectory but selectively enhanced the relative abundance of this genus > 10-fold (Fig. 7A, C; Supplementary Fig. 6a). The WD$_{McB}$-induced change in *Parabacteroides* abundance shifted the microbiota consortium away from their WD$_{REF}$-/LFD-fed counterparts (Fig. 7D, G). Considering the marked $_p$T$_{reg}$ phenotype, we next evaluated microbial structures in RAG2$^{-/-}$ mice deficient in adaptive immunity. Here, mice on LFD (Week 0) exhibited diminished *Parabacteroides* abundance as compared to their WT counterparts (Fig. 7A–C, E). Strikingly, WD$_{McB}$ feeding was incapable of boosting the relative abundance in RAG2 deficient mice, indicating that the selective increase reported above was mediated by WD$_{McB}$ through adaptive immunity, hence potentially by $_p$T$_{reg}$ induction (Fig. 7B, E). The reciprocal actions on *Desulfovibrio*

abundance was on the contrary *independent* of adaptive immunity (Fig. 7A, B; Supplementary Fig. 6a, b).

Despite some synergies between McB and T/B cells in affecting specific gut microbes (i.e., *Parabacteroides*), the global changes to microbial community structures induced by WD$_{REF}$-feeding in both WT and RAG2$^{-/-}$ mice was generally mitigated by WD$_{McB}$-feeding. These traits include a diminished F/B ratio in WDMcB-fed RAG2$^{-/-}$ mice (Supplementary Fig. 6c). Collectively, these findings suggest that the vast majority of WD$_{REF}$-induced changes and the McB-mediated protection against these, were independent of adaptive immunity (Fig. 7A–H; Supplementary Fig. 6a, b).

**Altered gut microbiota by WD feeding affects glucose regulation.** Despite a general protection against WD-induced global changes in the gut microbiota, the specific and substantial changes in key driver species (e.g., *Parabacteroides*) by WD$_{McB}$ feeding solely in WT mice, could suggest that some of the metabolic effects reported above was mediated by the selective increase in $_p$T$_{regs}$. We thus evaluated glucoregulatory capacity in RAG2$^{-/-}$ mice fed either diet for 6 weeks and observed a partial protection against WD-induced 5 h fasted hyperinsulinemia, insulin secretion during OGTT, body weight gain, and total fat mass (Fig. 8A–G). Both cecum weight and SCFAs were further enhanced to levels resembling those in the above-reported WT mice (Fig. 8H, I). Collectively, these data suggests that at least some of the metabolic effects observed in WD$_{McB}$-fed mice may occur independent of adaptive immunity.

We therefore designed CMT experiments in ABX treated mice and assessed the glucoregulatory capacity in cohorts fed either LFD or WD$_{REF}$ (Supplementary Fig. 1e). While we failed to observe a donor-dependent effect of CMT in LFD-fed mice (Fig. 8J, K; Supplementary Fig. 6d–g), we did observe transient tendencies of metabolic improvements, notably on fat mass and GSIS, in WD-fed mice receiving cecal microbes from LFD- or WD$_{McB}$-fed donors (Fig. 8K–O).

## Discussion

In this report, we explored the relationship between dietary nutrients and host-microbe interactions with a focus on immunometabolic response rates in the context of high fat, high sucrose, WD feeding. We found whole-cell lysates from the noncommensal methanotrophic bacterium, McB, capable of reversing hallmark signatures of WD feeding despite continued

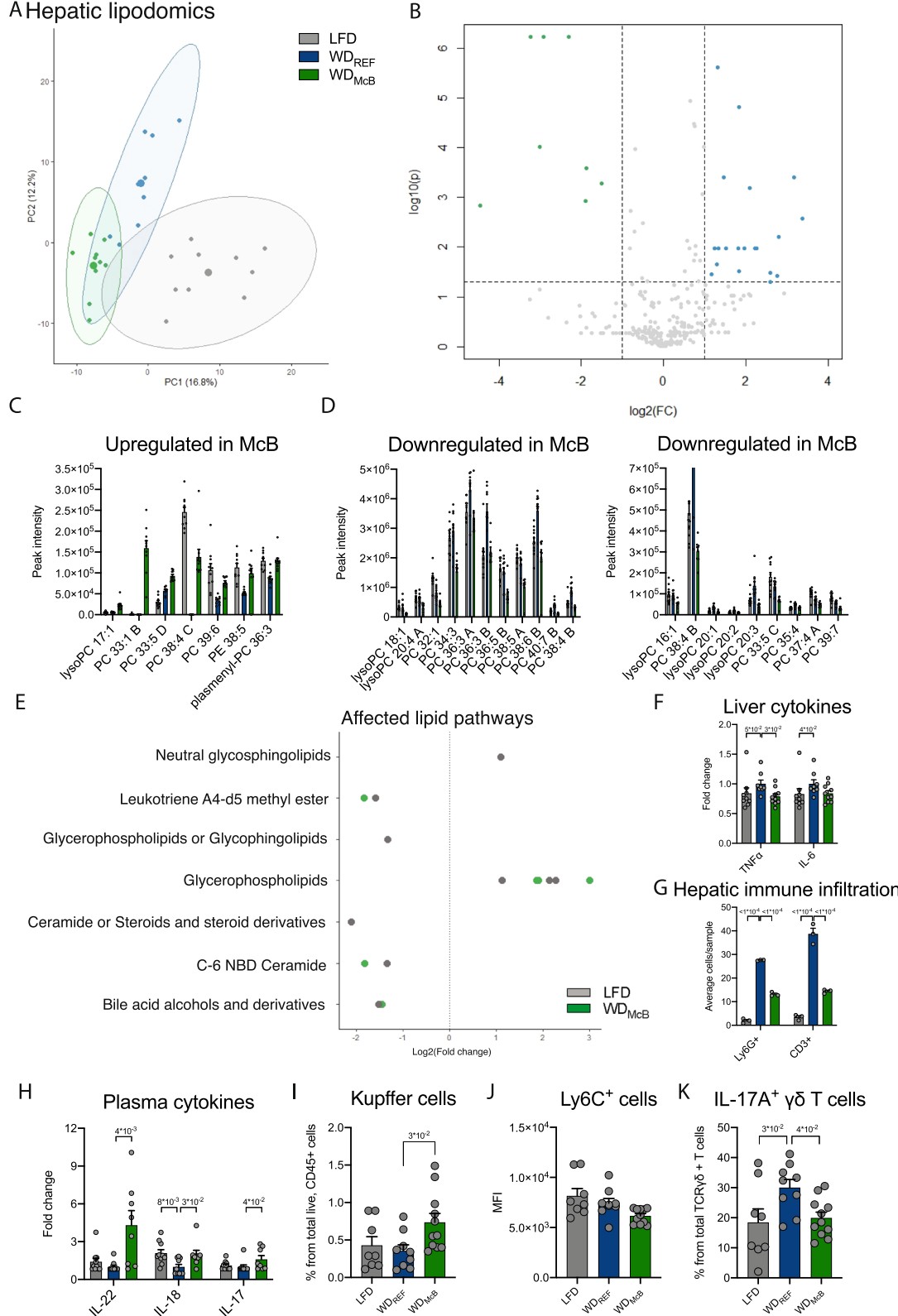

intake of an obesogenic diet. The signatures included diminished fat mass, improved intestinal immunity and glucoregulatory capacity accompanied by gut microbial community structures resembling that of lean LFD-fed counterparts.

The aberrant microbiota composition associated with obesity was recently shown to reflect HFD intake rather than obesity per se[45]. It is therefore worth noting that we were able to

normalize the dysregulated gut microbiota in mice remaining on a westernized, high fat, high sucrose diet; especially considering that it remains a clinical challenge to change dietary habits in individuals with lifestyle-related obesity. This remarkable change was reproduced in four substrains of mice originating from two different vendors, corroborating a robust phenotype not notably affected by baseline microbiota composition. To this end, a

**Fig. 5 WD$_{McB}$ feeding resets the hepatic lipidome and decreases hepatic immune infiltration alleviating NAFLD. A** Principle coordinate analysis (PCoA) of hepatic lipid species identified in negative ionization mode. **B** As in A but depicted in a volcano plot with significantly regulated lipid species (FDR < 0.05, adjusted for multiple comparisons by Benjamin–Hochberg) presented on a log2 scale in either green (WD$_{McB}$ > WD$_{REF}$) or blue (WD$_{McB}$ < WD$_{REF}$). **C, D** Identified lipid species differentially expressed (FDR < 0.01) between WD$_{McB}$ and WD$_{REF}$ as indicated. Adjusted $p$ values of individual lipid species are indicated in Supplementary Table 4. $n = 10$ (LFD and WD$_{McB}$) or 9 (WD$_{REF}$). **E** Lipid pathways identified by fold-change analysis (FDR < 0.05, adjusted for multiple comparisons by Benjamin–Hochberg) seperating LFD (gray) and WD$_{McB}$ (green) groups from WD$_{REF}$ in negative ionization mode. **F** Hepatic cytokine levels of TNF-α and IL-6 as indicated. $n = 9$ (LFD and WD$_{McB}$) and 8 (WD$_{REF}$). **G** Ly6G$^+$ and CD3$^+$ immune cells revealed by immunohistochemistry· $n = 3$ per group (randomly selected samples). **H** Cytokine levels in plasma of IL-22, IL-18, and IL-17. $n = 10$ (LFD and WD$_{REF}$) or 8 (WD$_{McB}$). **I** Amount of Tim4$^+$ Kupffer cells. $n = 8$ (LFD), 9 (WD$_{REF}$), and 11 (WD$_{McB}$) **J** Mean Fluorescence intensity (MFI) of Ly6C$^+$ liver monocytes. $n = 8$ (LFD), 9 (WD$_{REF}$), and 11 (WD$_{McB}$). **K** Amount of hepatic IL-17$^+$ γδ T cells. $n = 8$ (LFD), 9 (WD$_{REF}$), and 11 (WD$_{McB}$). **C, D, F–K** Bars represent group mean ± SEM and dots indicate individual data points. **F, H** Statistical significance compared to WD$_{REF}$ group by Kruskal–Wallis test, adjusted for multiple comparisons by Dunn's post-hoc. **G, I–K** Statistical significance compared to WD$_{REF}$ by one-way ANOVA, adjusted for multiple comparisons by Dunnett post-hoc. All $p$-values $< 1 \times 10^{-1}$ between WD$_{REF}$ and indicated group are depicted.

previous report on the resilience of the microbiota, argued for prolonged normalization[46]. This study found that mice transferred to a low fat, fiber-rich chow diet after 12 weeks of HFD feeding, shifted their microbiota towards age-matched chow-fed control mice within 4 weeks, but only fully converged 10 weeks post diet change. This is particularly interesting as dietary fibers are known to be the most potent dietary regulator of the gut microbiota[47], by far exceeding that of dietary fat[48]. Still, in our hands, WD$_{McB}$-feeding was able to reverse the obese microbiota at a higher pace than chow diet was in the previous report[46], despite similar fiber content in the two WDs. We further showed that the reversal of the microbiota traits was reproducible at different temperatures, altered reference diets, changed experimental duration, and in both cohoused and single-housed mice; all of which are prominent modulators of gut microbiota community structures. While many genera were similarly affected between experiments, others were either exclusively regulated at T$_{30°C}$ (e.g., *Bifidobacterium*) or less pronounced affected at T$_{30°C}$ (e.g., *Barnesiella*), suggesting dispensability for these specific microbes in the metabolic disease traits observed in this model. Similar observations were made for *Akkermansia*, which was ~3-fold upregulated at T$_{30°C}$, but discordantly affected (log2 FC bouncing from +9.3 to −10.4) between the two replication studies at T$_{22°C}$ with pronounced intragroup variation.

Contrasting these discrepancies, there was a consistent downregulation of *Desulfovibrio* accompanied by >10-fold upregulation of *Parabacteroides* and *Parasutterella* in immunocompetent mice when changed from either WD$_{CNTL}$ or WD$_{REF}$ to WD$_{McB}$ feeding. Our prophylactic experiment further showcased how *Parabacteroides* remained stable in LFD-, decreased in WD$_{REF}$- and dramatically increased in WD$_{McB}$-fed mice (~30-fold higher relative abundance than WD$_{REF}$-fed counterparts). *Parabacteroides* blooms were exclusively regulated in immunocompetent mice, suggesting that WD$_{McB}$ feeding promotes selective microbial traits by McB-T/B cell interactions, and this likely through the pronounced induction of IL-10$^+$/IL-17$^+$ gut-specific $_p$T$_{regs}$, being the only investigated cell subset that was notably changed upon WD$_{McB}$ feeding. It is worth noting, that not only was the prevalence of *Parabacteroides* not different between all groups and time points in RAG2$^{-/-}$ fed mice, the relative abundance of this genus in these mice was also lowered to a level resembling that of WD$_{REF}$-fed WT mice, hence pointing towards a strong involvement of adaptive immunity to support a *Parabacteroides* favorable ecological niche. Other commensals, such as *Barnesiella*, *Allobaculum*, *Clostridium* IV, and *Desulfovibrio* remained as *Parabacteroides* stable over time in LFD-fed mice, but were substantially and inversely regulated between WD$_{REF}$- and WD$_{McB}$-fed mice (Supplementary Fig. 6a, b). The feed and time-dependent trajectories of these specific bacteria were independent of adaptive immunity and thus similarly

regulated in both WT and RAG2$^{-/-}$ mice. Interestingly, *Desulfovibrio* was most recently shown to flourish in aged, immuno-compromised, obese mice with impaired glucose regulation[21]. Both T cell loss and *Desulfovibrio* administration per se, precipitated obesity-induced insulin resistance[21]. Still, in our hands, *Desulfovibrio* abundances neither were notably affected by the absence of T cells in RAG2$^{-/-}$ mice, nor were the suppressive capacity of our McB lysate. This bacterium has moreover been shown to thrive in desulfonated colonic mucosa, associated with human[49] and mouse[50] colitis. Combined, these observations lend credence to the hypothesis of a mechanistic link between WD$_{McB}$-mediated reductions of *Desulfovibrio* and the observed improvements in mucin chemotype and metabolic response rates. In support of this notion, we observed a partial protection against WD-induced metabolic impairments in WD$_{McB}$-fed RAG2$^{-/-}$ mice concomitant with *Desulfovibrio* suppression. Further pointing toward a role for the McB-induced microbiota changes to modulate host metabolism, WD$_{REF}$-fed, ABX treated recipient mice in CMT studies temporally exhibited a phenotype visually resembling that of their donors. Despite following a protective trajectory, the phenotype remained insignificant, likely due to a combination of the inherent variation in such experiments, a rather low n-number based on sample availability, and only a partial involvement of the gut microbial community structure in the precipitated phenotype.

Future studies are warranted to elucidate host-microbe mutualism upon WD$_{McB}$ feeding. On this note, mucin-producing goblet cells are capable of delivering luminal antigens to LP-residing DC[51] instrumental for T cell polarization[52]. As McB potently induce DC maturation markers in vitro—even exceeding the effects of the well-described probiotic strain, *Escherichia coli* Nissle 1917[53]—affecting subsequent T cell reponses[16], we predict a direct link between McB intake and the corresponding immune profile, which might subsequently precision edit certain microbes, e.g., *Parabacteroides*.

In addition to the direct link proposed here, McB intake might also indirectly promote T$_{reg}$ polarization through augmented SCFAs[54], a trait persisting in RAG2$^{-/-}$ mice. These important metabolites also sustain mucus production and facilitate tissue crosstalk in the gut-liver axis[2]. In keeping with this notion, we observed decreased hepatic bile acids and TNF-α levels combined with a pronounced reduction of intrahepatic CD3$^+$ and Ly6G$^+$ cells in obese WD$_{McB}$-fed mice. Importantly, while Ly6C$^{high}$ cells represent proinflammatory, fibrogenic macrophages[55], their in situ differentiation to Ly6C$^{low}$ cells[41] facilitate tissue repair and improved NASH prognosis[42]. It is therefore encouraging to note, that the MFI of Ly6C was nominally decreased in 'weight-matched' mice fed WD$_{McB}$ short term. These data indicate that, despite being increased in absolute numbers, Ly6C$^+$ cells in WD$_{McB}$-fed mice were in the process of differentiating from

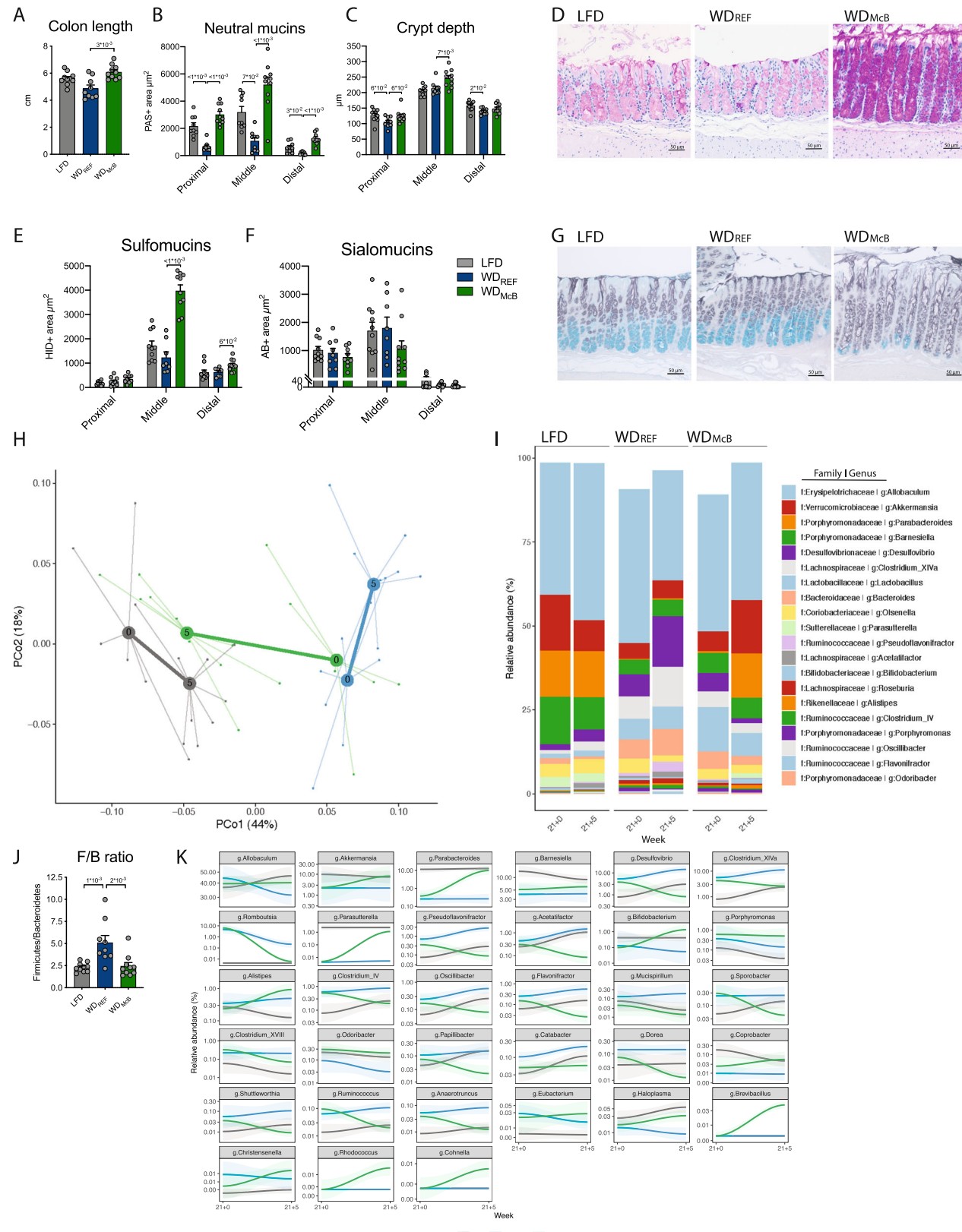

Ly6C$^{high}$ to Ly6C$^{low}$ cells with immune resolving capabilities, although further studies are warranted to fully describe the kinetics in such responses. In agreement with an immune resolving phenotype, WD$_{McB}$-fed mice were fully protected from the ~50% increase in hepatic IL-17$^{+}$ γδ T cells observed in WD$_{REF}$-fed counterparts before the onset of obesity. This cell subset has been shown to (a) be regulated by the gut microbiota and (b) both precede and causally induce NAFLD[43]. The above affected targets are all proposed as relevant strategies to curb NAFLD and the more terminal liver disease, NASH[36].

The substantial, consistent, and global changes in the immunometabolic profile found in this study point towards clinical

**Fig. 6 WD$_{McB}$ treatment improves colonic mucus production and reverses obesity-induced gut microbiota changes to resemble the composition found in lean LFD-fed mice. A** Colon length in centimeters. Statistical significance compared to WD$_{REF}$ group by Kruskal–Wallis test, adjusted for multiple comparisons by Dunn's post-hoc. $n = 10$ (LFD and WD$_{McB}$) or 9 (WD$_{REF}$). **B** Area of neutral mucins determined by Periodic Acid Schiffs (PAS) staining in three discrete locations, i.e., proximal, middle, and distal colon segments. Three longitudinally sectioned crypts were measured in each sample (one data point) at each location. Statistical significance compared to WD$_{REF}$ group by one-way ANOVA, adjusted for multiple comparisons by Dunnett post-hoc, for proximal and distal segments while middle segment was by Kruskal–Wallis test, adjusted for multiple comparisons by Dunn's post-hoc. $n = 9$ (LFD and WD$_{REF}$) or 10 (WD$_{McB}$) except for distal segment where $n = 9$ for WD$_{McB}$. **C** Colon crypt depth (CD) in proximal, middle, and distal colon segments, measured in histological sections. Measurements are given in μm, and results are given as means of six crypt measurements per location per individual mice (3 longitudinally PAS stained and HID-AB stained crypts, respectively, per data point). Statistical significance compared to WD$_{REF}$ group by one-way ANOVA, adjusted for multiple comparisons by Dunnett post-hoc test. $n = 10$ (LFD and WD$_{McB}$) or 9 (WD$_{REF}$). **D** PAS staining for mucus in colon middle segments. One representative picture per group, as indicated, out of 9 (LFD and WD$_{REF}$) or 10 (WD$_{McB}$) middle segment colon sections. **E** Area of sulfomucins in proximal, middle, and distal colon segments by HID staining. Three longitudinally sectioned crypts were measured in each sample and at each location. Statistical significance compared to WD$_{REF}$ group by one-way ANOVA, adjusted for multiple comparisons by Dunnett post-hoc, for proximal and middle segments while distal segment was by Kruskal–Wallis test, adjusted for multiple comparisons by Dunn's post-hoc. LFD $n = 10$, WD$_{REF}$ $n = 8$ except proximal segment where $n = 9$, and WD$_{McB}$ $n = 10$ except for proximal segment where $n = 9$. **F** Area of sialomucins in proximal, middle, and distal colon segments by AB staining. Three longitudinally sectioned crypts were measured in each sample and at each location. Statistical significance compared to WD$_{REF}$ group by one-way ANOVA, adjusted for multiple comparisons by Dunnett post-hoc, for proximal and middle segments while distal segment was by Kruskal–Wallis test, adjusted for multiple comparisons by Dunn's post-hoc. LFD $n = 10$, WD$_{REF}$ $n = 8$ except proximal segment where $n = 9$, and WD$_{McB}$ $n = 9$ except for middle segment where $n = 10$. **G** Representative HID-AB staining of middle segment of indicated experimental group. One representative picture per group out of 10 (LFD and WD$_{McB}$) or 8 (WD$_{REF}$) evaluated sections. **H** PCoA of fecal microbiota composition of indicated group before (Week $21 + 0$) and after ($21 + 5$) dietary intervention with group mean indicated as centroids. Microbiota composition was significantly different between the WD$_{REF}$ and WD$_{McB}$ groups at the end of the experiment week $21 + 5$ (PERMANOVA $p = 0.001$). **I** Taxasummary of most abundant bacterial genera showing mean relative abundance of indicated genera in each group at indicated timepoint. **J** Firmicutes/Bacteroidetes ratio of fecal samples of individual mice at termination (Week $21 + 5$). Statistical significance compared to WD$_{REF}$ group by one-way ANOVA, adjusted for multiple comparisons by Dunnett post-hoc test. $n = 10$ (LFD and WD$_{McB}$) or 9 (WD$_{REF}$). **K** Deseq analysis of fecal bacterial genera abundances significantly regulated by McB intervention (p.adj. < 0.05). Relative abundance in % in each group and variation are shown for each regulated genera at the sampled time points. Fold-change and adjusted $p$ values of individual genera are indicated in Supplementary Table 5. **A–K** $n = 8$–10. **A–C, E, F, J**) Bars represent group mean ± SEM and dots indicate individual data points. All $p$-values $< 1 \times 10^{-1}$ between WD$_{REF}$ and indicated group are depicted.

potential if further developed. Notably, only a single study has previously succeeded in inducing $_p$T$_{regs}$, and this was limited to LI-LP[11]. Apart from inducing this cell subset throughout the gastrointestinal tract, we further show that these cells exhibited enhanced secretory capacity of both the hallmark immunosuppressive cytokine, IL-10, as well as IL-17 in LI-LP of WD$_{McB}$-fed mice compared to their WD$_{CNTL}$ fed counterparts. To this end, emerging evidence indicates that the non-regulatory counterparts to $_p$T$_{regs}$, i.e. T$_H$17 cells, are purged from SI-LP in DIO mice[14] and that reintroducing ex vivo differentiated gut tropic T$_H$17 cells curtail obesity development, hence improving insulin sensitivity[15]. Notably, despite intergroup similarities in the relative proportion of SI-LP IL-10$^+$/IL-17$^+$ $_p$T$_{regs}$, the absolute numbers of these cells were markedly increased in WD$_{McB}$-fed mice, thus advocating enhanced immune regulation.

That IL-17 in this intervention is produced by T$_{regs}$ and not T$_H$17 cells might further improve the safety profile of a putative medication, as the physiological impact and properties of T$_H$17 cells are context-dependent. As such, T$_H$17 cells may potentiate inflammatory bowel disease (IBD) in individuals with compromised barrier function[56], whereas $_p$T$_{regs}$ have shown enhanced suppressive capacity against the same disease[12]. This is a pertinent feature given the relatively high proportion of subjects with metabolic complications co-suffering from IBD[57]. While our study was not designed to assess IBD susceptibility, it is well described that purified diets compromise barrier function[47,58,59] and thus represent clinical IBD-features (disease-cause) preceding inflammation (symptom)[60]. In addition to the immunomodulating phenotype, WD$_{McB}$-feeding augmented neutral mucus production in all segments of the colon while enhancing IBD-protective sulfomucins specifically in the middle segment. In this segment, we also detected increased CD in WD$_{McB}$-fed mice, collectively pointing towards increased barrier function.

$_p$T$_{regs}$ were most recently described to be transferred from mother to offspring via nongenetic inheritance through breastmilk IgA; a trait transmitted during a tight age window after birth but stable for life, and resistant to many microbial or cellular perturbations[61]. It is thus striking to note the pronounced and reproducible induction we here report in both SI- and LI-LP in three substrains of mice from different vendors. Unfortunately, we have not investigated the B cell component of the immune system in this study, and thus, we cannot conclude on the extent to which $_p$T$_{reg}$ induction relies on B cells, McB lysate per se or the McB-induced SCFAs and/or gut microbial changes. It is, however, worth noting that apart from IgA-induced $_p$T$_{reg}$-control in early life, other reports have also convincingly demonstrated that this cell subset can be induced by bacterial metabolism of bile acids[62,63] as well as microbe-derived SCFAs in general[54] and butyrate in particular[64]. The latter of which we found abundantly increased upon WD$_{McB}$ feeding.

In summary, this study demonstrates a consistent activation of IL-10$^+$/IL-17$^+$ $_p$T$_{regs}$ throughout the gastrointestinal tract and a reversion of gut dysbiosis in response to WD$_{McB}$ feeding, even in the context of high dietary fat and sucrose intake. The prospects of using bacterial lysates as an alternative to traditional live probiotics merit further investigation, just as future studies are warranted to assess to what extent our results can be translated to humans. Future studies are also urgently needed to identify the biological effector molecule(s) in this bacterial lysate, with a potential for being developed into a medical product(s) rebalancing intestinal immunity while targeting gut-related dysbiosis and metabolic abnormalities.

## Methods

**Mice and ethical statements.** All experiments were conducted in accordance with the EU directive 2010/63/EU as approved by the Danish Animal Experiments Inspectorate (#2014-15-2934-01,027). Six- to seven-week-old male C57BL/6JBomTac, C57BL/6JRj, C57BL/6 N, C57BL/6N-Rag2Tm1/CipheRj (RAG2$^{-/-}$) mice were acquired from Taconic Laboratories, Denmark, or Janvier Labs, France, respectively, as detailed below. All mice were allowed to acclimatize on regular chow diet for the first week upon arrival and then fed a low fat diet (LFD) for

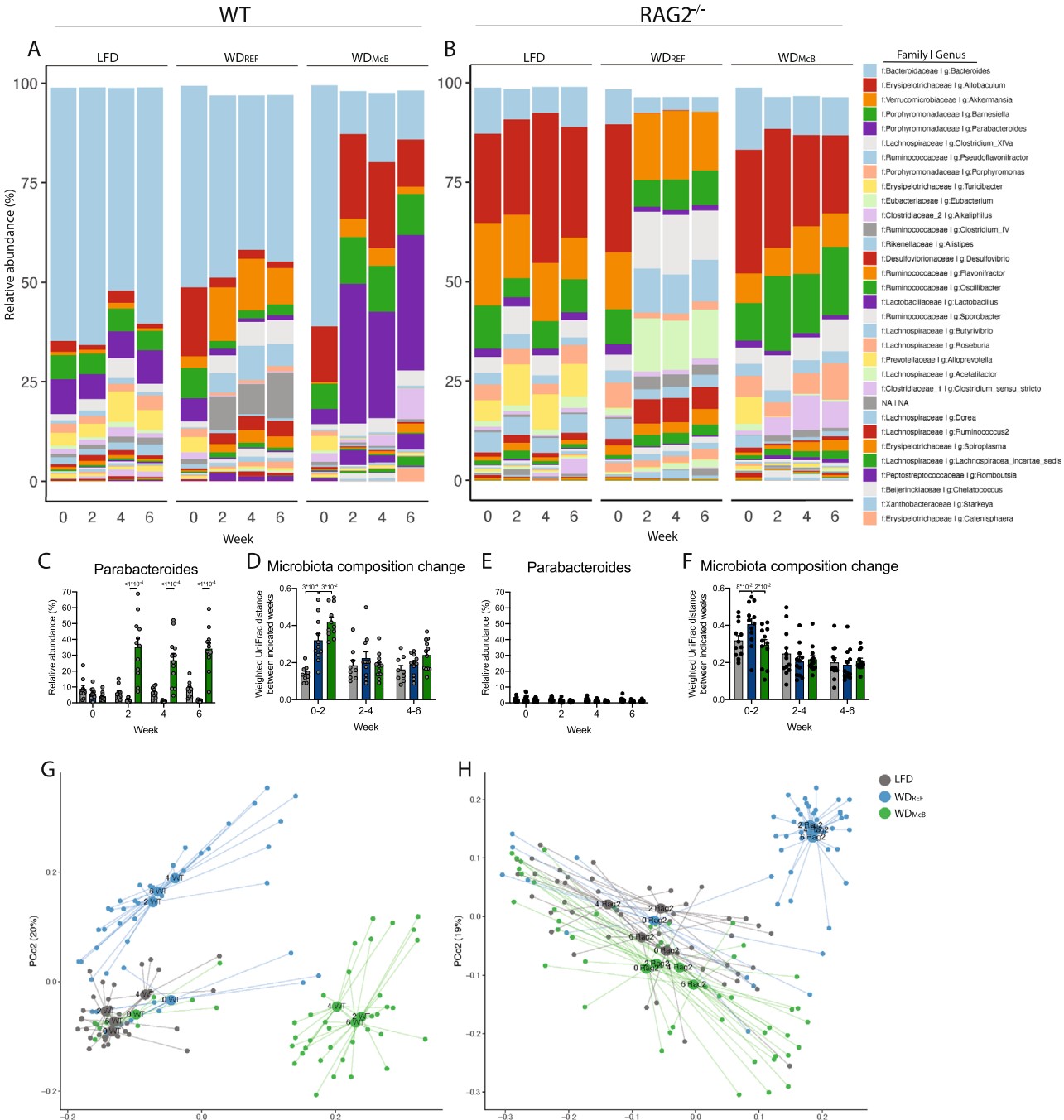

**Fig. 7 McB lysates rely on adaptive immunity to boost *Parabacteroides*. A, B** Taxasummary of most abundant bacterial genera showing mean relative abundance of indicated genera in each group at indicated timepoint in WT (**A**) and RAG2$^{-/-}$ (**B**) mice. **C, E** Relative abundance of *Parabacteroides* in fecal samples of indicated genotype at indicated timepoint. **D, F** Weighted UniFrac distance (instability test) between paired samples from indicated 2-weeks interval post dietary intervention in WT (**D**) and RAG2$^{-/-}$ (**F**) mice. **C–F** Bars represent group mean ± SEM and dots indicate individual data points. All *p*-values < 1 × 10$^{-1}$ between WD$_{REF}$ and indicated group are depicted. Statistical significance compared to WD$_{REF}$ by two-way ANOVA-RM, adjusted for multiple comparisons by Dunnett post-hoc. **G, H** PCoA of fecal microbiota composition of indicated group at baseline (Week 0, all mice fed LFD) and after 2, 4, and 6 weeks of dietary intervention with group mean indicated as centroids.

2 weeks prior to study initiation. Mice were kept under specific pathogen free conditions at 22 °C (T$_{22°C}$) or 30 °C (T$_{30°C}$), as indicated, in 12 h light/dark cycle (6AM–6PM).

**Housing, diets, and experimental setup**. Standard pelleted diets as well as protein-free western diet (WD) powder were obtained from Ssniff Spezialdiäten GmBH, Germany, and stored at −20 °C throughout the duration of the experiment. Mice were fed a compositionally defined low fat diet (LFD, S8672-E050) until study initiation. A subgroup remained on LFD, whereas the remaining mice

were transferred to a soy oil based reference WD (S8672-E025) (high fat, high sucrose, containing 0.15% cholesterol) for the run-in period (12 or 21 weeks dependent on the experiment). Experimental WDs were lot-matched and produced without protein (S9552-E021), which was subsequently added, and then pelleted, by the investigators as indicated; WD$_{REF}$ = containing 19.5% (w/w) casein, WD$_{CNTL}$ = containing 16.5% (w/w) casein and 3% Macadamia oil, WD$_{McB}$ = 19.5% (w/w) whole-cell bacterial lysates (predominantly protein but up to ~15% phospholipids[31]). Diet composition is more extensively described in Supplementary Table 1 with amino acid composition presented in Supplemetary Fig. 1f. Detailed description of bacterial lysates is provided in the designated section below.

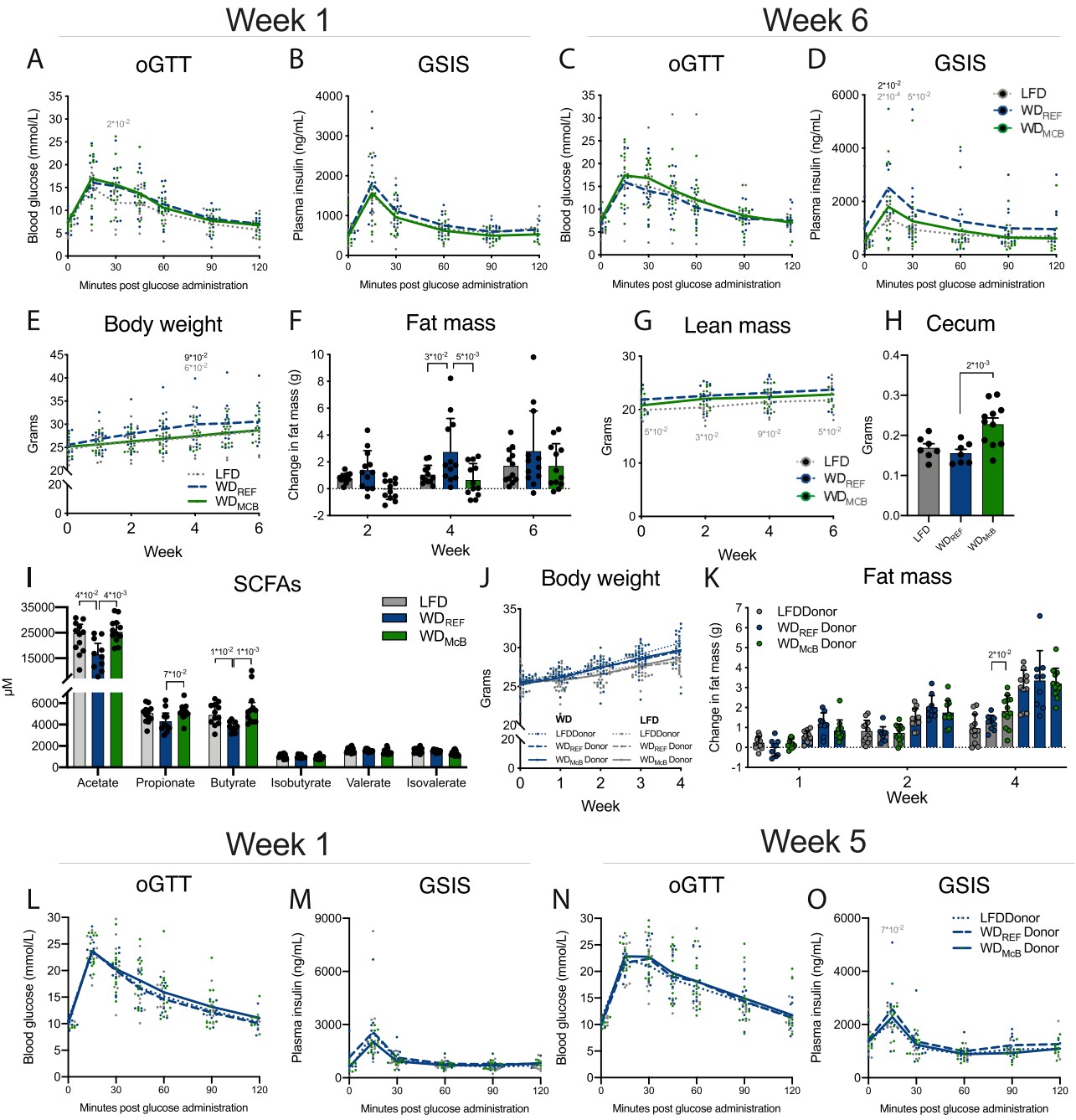

All diets were freshly made for each experiment with independent batches of bacterial lysates to substantiate robustness and reproducibility of the observed findings. Mice were fed ad libitum, and feed intake was measured thrice a week. Water was changed weekly.

*12 + 6 protocol $T_{22°C}$.* Experimental setup is outlined in Supplementary Fig. 1a. C57BL/6JRj mice were purchased from Janvier Labs, France, and housed three mice per cage, reproduced in two independent experiments. For transparency, all data are depicted in respective graphs with mean ± SEM plus individual dots for the combined results of the two independent experiments. Experiment-dependent shapes enable visual discrimination of respective experiments as well as data distribution within and between experiments.

*21 + 5 protocol $T_{30°C}$.* Experimental setup is outlined in Supplementary Fig. 1b. C57BL/6JBomTac mice were purchased from Taconic Laboratories, Denmark, and initially housed 10 per cage to accelerate weight gain. Twelve weeks into the run-in period, mice were single caged to allow accurate assessment of food intake, and also monitor if the more sensitive microbiota of single-housed mice would uniformly change towards a composition typical for LFD-fed mice, or if such phenomenon relied on a few highly responding mice transferring their microbiota to cage mates

under group housed conditions. The timing of single housing allowed affected mice to adapt to social isolation and stabilize their weight development before intervention (Supplementary Fig. 5b).

*Prophylactic protocols.* Experimental setup is outlined in Supplementary Fig. 1c, d. C57BL/6N-Rag2Tm1/CipheRj (RAG2$^{-/-}$ mice) and matched C57BL6/N WT mice were purchased from Janvier Labs, France, and housed three mice per cage at thermoneutrality, acclimated on LFD and from 9 weeks of age fed the experimental diets for 6–7 weeks depending on genotype (see Supplementary Fig. 1c, d for details). This was reproduced in two independent experiments analogous to the 12 + 6 protocol described above.

*Cecal microbiota transfer (CMT) protocol.* Experimental setup is outlined in Supplementary Fig. 1e. C57BL/6JBomTac mice were purchased from Taconic Laboratories, Denmark, and housed three mice per cage at thermoneutrality, reproduced in two independent experiments. Experiment-dependent shapes enable visual discrimination of respective experiments as well as data distribution within and between experiments. We used mice from the 21 + 5 protocol as donors of their cecal microbiota, ensuring transfer between genetically identical mice hence limiting obvious confounders. Cecal microbiota transfer (CMT) was performed as

**Fig. 8 Altered gut microbiota by WD feeding affects glucose regulation. A, C** OGTT in 5 h fasted RAG2$^{-/-}$ mice after 1 and 6 weeks of dietary intervention as indicated. Statistical significance compared to WD$_{REF}$ by two-way ANOVA-RM, adjusted for multiple comparisons by Dunnett post-hoc. $n = 12$ mice per group. **B, D** Glucose stimulated Insulin Secretion during OGTT after 1 and 6 weeks of dietary intervention. Statistical significance compared to WD$_{REF}$ by mixed effects analysis RM, adjusted for multiple comparisons by Dunnett post-hoc. $n = 12$ mice per group except for WD$_{McB}$ group at 120 min in **B**, WD$_{McB}$ group at 60 min in **D**, and WD$_{REF}$ group at 15 min in **D** where $n = 11$ due to insufficient sample material. **E–G** Body weight, fat mass and lean mass, as indicated, at indicated time points post dietary intervention. Statistical significance of body weight compared to WD$_{REF}$ by mixed effects analysis RM, adjusted for multiple comparisons by Dunnett post-hoc, and fat and lean mass by two-way ANOVA-RM, adjusted for multiple comparisons by Dunnett post-hoc. $n = 12$ mice per group except for WD$_{McB}$ group Week 1 in **E** where $n = 9$. **H** Cecum weight at termination. Statistical significance compared to WD$_{REF}$ group by one-way ANOVA, adjusted for multiple comparisons by Dunnett post-hoc test. $n = 7$ (LFD and WD$_{REF}$) or 11 (WD$_{McB}$). **I** Concentration of short chain fatty acids in cecum from **H**. Statistical significance compared to WD$_{REF}$ group by Kruskal–Wallis test, adjusted for multiple comparisons by Dunn's post-hoc. $n = 12$ (LFD and WD$_{McB}$) or 10 (WD$_{REF}$). **J, K** Body weight and fat mass development in ABX treated WT mice fed either LFD (grey bars) or WD$_{REF}$ (blue bars), receiving cecal microbiota from indicated donor mice. Statistical significance compared to WD$_{REF}$ by two-way ANOVA-RM, adjusted for multiple comparisons by Dunnett post-hoc. WD-fed mice $n = 11$ (LFD donor), 9 (WD$_{REF}$ donor), and 12 (WD$_{McB}$ donor). LFD-fed mice $n = 12$ (LFD donor), 9 (WD$_{REF}$ donor), and 12 (WD$_{McB}$ donor). **L, N** OGTT in 5 h fasted WD$_{REF}$-fed WT recipient mice 1 and 5 weeks after first cecal microbiota transfer, as indicated. Statistical significance compared to WD$_{REF}$ by two-way ANOVA-RM, adjusted for multiple comparisons by Dunnett post-hoc. LFD donor $n = 11$, WD$_{REF}$ donor $n = 9$, and WD$_{McB}$ donor $n = 12$. **M, O** Glucose stimulated Insulin Secretion during OGTT (L + N) 1 and 5 weeks after first cecal microbiota transfer. Statistical significance compared to WD$_{REF}$ by two-way ANOVA-RM, adjusted for multiple comparisons by Dunnett post-hoc. In M, LFD donor $n = 11$ except at 0 min where $n = 4$, WD$_{REF}$ donor $n = 9$ except at 0 min where $n = 6$, and WD$_{McB}$ donor $n = 12$ except at 0 min where $n = 9$. In O, LFD donor $n = 10$, WD$_{REF}$ donor $n = 8$ except at 90 min where $n = 9$, and WD$_{McB}$ donor $n = 11$ except at 90 min where $n = 12$. Reduced n-size reflects insufficient sample material at indicated timepoint. **A–E, G, J, L–O** Lines represent group mean and dots indicate individual data points. **F, H, I, K** Bars represents mean ± SEM. Dots indicate individual data points. **A–O** All p-values $< 1 \times 10^{-1}$ between WD$_{REF}$ and indicated group are depicted.

---

after one week of antibiotic (ABX) treatment. Specifically, mice were housed in disposable IVC cages with free access to sterilized water and LFD during the 2 weeks of acclimatization. They were then treated with a two-leg ABX mixture (1 and 0.5 g ampicillin and neomycin, respectively, per liter water, changed thrice weekly) for seven days while remaining on LFD. After ABX treatment, experimental mice were fasted for 2 h and subsequently gavaged with Polyethylene Glycol (Gangiden, Sandoz) to clean the intestines from potential remnants. Six hours later, mice received the first of three boluses of cecal microbiota from either LFD-, WD$_{REF}$-, or WD$_{McB}$-fed donors. Cecal microbiota were pooled from all mice of the respective groups before transfer. Immediately after first CMT, mice were divided into either LFD- or WD$_{REF}$-fed recipients and fed their respective diets for additionally 5 weeks.

**Cultivation of *M. capsulatus* Bath and preparation of bacterial lysate.** *Methylococcus capsulatus* Bath (NCIMB 11132) was cultivated in nitrate mineral salts (NMS) medium to produce the single-strain bacterial lysate. NMS medium was composed from 5 stock solutions; 10× NMS Salts (98.9 mM KNO$_3$, 43.8 mM MgSO$_4$*6H$_2$O and 9.0 mM CaCl$_2$), 1000× NaMoO$_4$*2H$_2$0 (1.07 mM), 10,000× FeEDTA (103 mM), 1000× Trace Elements Solution (0.8 mM CuSO$_4$*5H$_2$0, 1.8 mM FeSO$_4$*7H$_2$0, 1.4 mM ZnSO$_4$*7H$_2$0, 0.24 mM H$_3$BO$_3$, 0.21 mM CoCl$_2$*6H$_2$0, 0.74 mM EDTA-Na$_2$, 0.1 mM MnCl$_2$*4H$_2$0, 42.07 uM NiCl$_2$*6H$_2$0) and 10× phosphate buffer pH 6.8 (198.25 mM Na$_2$HPO$_4$*12H$_2$0, 191.05 mM KH$_2$PO$_4$). When preparing 1× NMS medium from stock solutions 10x NMS Salts was add to 50% of the final volume with H$_2$O before the remaining stock solutions were added. Then pH was adjusted to 6.8 before the NMS medium was sterile filtered using a 0.45-uM filter. Medium was stored in the dark.

The 1000× Trace Element Solution was made from the following stock solutions and stored in the dark; 400.5 mM CuSO$_4$*5H$_2$0, 3.6 mM FeSO$_4$*7H$_2$0 (pH 3.0), 347.8 mM ZnSO$_4$*7H$_2$0, 161.7 mM H$_3$BO$_3$, 42.0 mM CoCl$_2$*6H$_2$0, 50.5 mM MnCl$_2$*4H$_2$0 and 42.0 mM NiCl$_2$*6H$_2$0. McB culture aliquots were frozen in liquid nitrogen and stored at −80 °C. Cultivations on agar plates and in shake flasks (orbital shaker incubator at 200 rpm) were performed at 45 °C in an atmosphere of 75% air, 23.25% CH$_4$, and 1.25% CO$_2$. Continuous cultivation was carried out in a 3-l bioreactor (Applikon, The Netherlands) with a working volume of 2 l. Cells were precultivated in shake flasks and used to inoculate the bioreactor to an optical density at 440 nm (OD$_{440}$) of 0.1. The temperature was maintained at 45 °C, stirring set to 650 rpm, and pH maintained at 6.8 by automatic addition of 2.5 M NaOH/2.5 M HCl. A gas mixture of 75% air and 25% methane was sparged into the bioreactor. The continuous culture was started after an initial batch phase, and the dilution rate was set to 0.01 h$^{-1}$. The OD$_{440}$ at steady state was generally sustained at ~10. Culture effluent was collected, and cells were harvested by centrifugation. Bacterial cell walls were disrupted by the use of a French press before freeze-drying of the material.

**Glucose and insulin tolerance tests.** Mice were subjected to magnetic resonance (MR)-scan using EchoMRI 4in1 (Texas, USA) to determine fat- and lean mass at indicated time points (Supplementary Fig. 1a–d). Mice were fasted 5 or 2 h prior to any oral glucose tolerance (OGTT) or intraperitoneal insulin tolerance test (ITT), respectively. Fasting blood glucose was measured by tail vein bleeding using the Bayer Contour glucometer (Bayer Health Care). Mice were subsequently gavaged with 3 µg glucose/g lean mass (OGTT) or intraperitoneally injected with 0.75 mU

insulin/g lean mass (ITT). Blood were sampled at specified time points for blood glucose and insulin measurements to assess glucoregulatory capacity including glucose-stimulated insulin secretion (GSIS) as described in detail elsewhere[28].

**Short chain fatty acid measurements.** Short chain fatty acids (SCFAs) were determined by gas chromatographic analysis. Feces were suspended in MilliQ water (1:1 (w/v)) and subsequently homogenized in a FastPrep-96 (MP Biomedicals) sample preparation unit. The homogenates were diluted with 0.4% formic acid (1:1 (v/v)), transferred to Eppendorf tubes and centrifuged at 16,200 rcf for 10 minutes. 300 µl of the supernatants were applied to spin columns (VWR, 0.2 µm pore size) and centrifuged at 9560 rcf for 5 minutes. The eluates were transferred to 300-µl GC vials. Split injection mode was used with an injection volume of 0.2 µl. The gas chromatograph was a Trace 1310 (Thermo Scientific) equipped with an auto-sampler and a flame ionization detector. Helium was used as carrier gas, and the column was a 30 m long Stabilwax (Restek) with polyethylene glycol as stationary phase. Injector temperature: 250 °C, temperature intervals: 2 min at 90 °C before a 6 min increase to 150 °C, then a 2 min increase to 245 °C and a hold at this temperature for 4.9 min. Detector temperature: 275 °C.

**RNA extraction and quantitative RT-PCR.** Frozen liver tissue was cryo-grinded by mortar and pestle on liquid nitrogen and total RNA was extracted by TRIr-eagent (Sigma–Aldrich, USA) according to the manufacturer's protocol using PRECELLYS® 24 for homogenization. One microgram of RNA was transcribed into cDNA by reverse transcriptase (Invitrogen, USA). Quantitative PCR analyses were performed using the SYBR Green qPCR Master mix (Thermo Scientific, USA) and the Stratagene Mx3000P qPCR System. Standard curve and no template control (NTC) were included on every 96-well plates. Validation of each target was based on Rsq of standard curve >0.995, amplification efficiency of 100 ± 10% and singular peak. Ct values for each target were related to 18 S reference Ct values of the same sample by 2$^{\Delta Ct}$ and visualized as fold-change to the LFD reference group mean. Primer sequences are summarized in Supplementary Table 2.

**Histology.** Liver tissue was sampled promptly after euthanasia, fixed in 10% formalin and subsequently embedded in paraffin prior to sectioning according to standard procedures for light microscopy. Colon was emptied for content and subsequently rolled around a 27 G needle after which the generated Swizz Rolls were carefully preserved in liquid nitrogen (wrapped in aluminum foil) until paraffin embedding. Embedded tissues were cut 2-µm thick, and sections mounted onto glass slides. These sections were processed further and stained either with hematoxylin and eosin (H&E), Oil Red O, PAS, or HID-AB, or labelled with antibody of interest for immunohistochemical investigations, as detailed below. Samples were randomized and blinded to the pathologists performing the histological analyses.

*Liver.* **Nonalcoholic fatty liver disease activity score (NAS).** Liver sections were prepared by standard protocols and subsequently assessed by two independent observers in a blinded fashion, using the established NAFLD activity score (NAS) for evaluation of H&E stained liver sections[65]. In short, score of 0–2 excludes nonalcoholic steatohepatitis (NASH), a score of 3–4 defines "borderline NASH", and score ≥5 is considered as NASH.

**Oil Red O staining**. Liver biopsies were fixed in paraformaldehyde, cryoprotected in polyvinylpyrrolidone/saccharose, and frozen in liquid nitrogen. Cryosections were prepared and stained using the lipid-specific Oil red O (Sigma–Aldrich, Germany) and analyzed under an Axio Imager M2 microscope. Images were captured with the Axiocam 506 color camera (Zeiss, Jena, Germany). Areas of stained lipids were determined using ImageJ-1.50i software (https://imagej.nih.gov/ij/index.html).

*Colon*. The morphological and morphometrical evaluations of the colonic tissues were performed in a blinded fashion, and all measurements were made by the same trained pathologist. One H&E stained section of colon from each individual was evaluated histologically for any sign of pathology. To assess the quality of the mucins, colonic tissues were stained for acidic as well as neutral mucins. Neutral mucins were stained by Periodic Acid Schiffs (PAS). Acidic mucins were stained through a combination of High Iron Diamine (HID), for sulfated mucins (sulfomucins), and Alcian Blue (AB) for carboxylated mucins (sialomucins). Crypt depth (CD) measurements, as well as analyses of mucin type and amount, were performed on HID/AB- and PAS-stained sections in three histologically distinct areas of the colon, i.e., the proximal, middle and distal area. At each of these three locations, three longitudinally sectioned crypts were measured. Each data point is thus presented as an average of three (mucin type and amount) or six (CD) measures. Crypts were selected only when the entire crypt epithelium was visible from the *lamina muscularis mucosae* to the lumen. The histological sections of the colon were examined in an Axio Imager Z2 microscope (Zeiss, Jena, Germany), and digital images were obtained using an Axiocam 506 color camera (Zeiss, Jena, Germany). Micrographs were captured with the same ×20 objective magnification. Crypt depth measurements were performed using the software program Image J-Fiji version 1.52e Java 1.8.0_181[66]. For quantitative evaluation and automated scoring of the different mucin types, a plugin for color deconvolution for the Image J-Fiji program was used[67].

**Hepatic lipidomics**. Samples were randomized and lipid extracted following standard protocols. In short, 50 mg of frozen tissue were homogenized in a pre-chilled 2 mL tube added 0.5 mL of ice-cold 50% MeOH with ~40 µM D5-tryptophan following standard protocols and stored at −80 °C until LCMS analysis. The peaklist obtained after preprocessing (features defined by their mass/charge value, retention time and peak area) was analyzed in MetaboAnalyst 4. Data were then auto-scaled (mean-centering and division by the square root of standard deviation of each variable) to enforce Gaussian distribution enabling relative comparison. Univariate and multivariate analysis were performed with t-tests, volcano plot, analyses of variance (ANOVA) followed by Tukey's HSD test, and principal component analysis (PCA) to detect significant hits and to visually separate trends between groups. Features showing a similar pattern between LFD and WD$_{McB}$ were identified by pattern matching function. The top 25 features obtained were then processed for identification using the online database Lipidmaps (http://lipidmaps.org) with a mass tolerance between the measured m/z value and the exact mass of 3 ppm.

**Microbiome analysis and bioinformatic processing**. Fresh feces samples were collected 3–4 h into the light cycle, snap frozen and stored at −80 °C until downstream processing. Bacterial DNA from fecal samples was extracted using a NucleoSpin soil kit (Macherey-Nagel) according to manufacturer's instructions. 16 S rRNA gene amplification was performed using 515 F and 806 R primers and sequenced with 250PE on Ilumina MiSeq. USEARCH[68] and mothur[69] were used for the processing of the sequence data. Sequences were strictly dereplicated, discarding clusters smaller than 5. Sequences were clustered at 97% sequence similarity. Additional suspected chimeric operational taxonomic units (OTUs) were discarded on the basis of comparison with Ribosomal Database Project classifier training set v9[17] using UCHIME[70]. Taxonomic assignment of OTUs was done using the database from Ribosomal Database Project[17]. Initial data processing was performed as described elsewhere[71]. Data was rarefied to 23,107 reads per sample. The principal coordinate analysis (PCoA) was conducted based on the weighted UniFrac distance. To select OTUs and species enriched in different subgroups, DESeq2[72] with default settings was used. The relative abundances of OTUs were aggregated to genus level. Genera present in >1/3 of the samples were used in the differential abundance testing. Genera with p.adj. < 0.05 comparing the last sampling point for WD$_{REF}$/WD$_{CNTL}$ vs WD$_{McB}$ with DESeq2 differential expression analysis based on the negative binomial distribution were reported. Permutational multivariate analysis of variance (permanova) using weighted UniFrac distance matrices (Adonis, Vegan R package) was used to evaluate overall microbial compositions.

**Isolation of small intestine (SI) and large intestine (LI) lamina propria (LP) cells**. Feces and mucus from LI were scraped off while SI was flushed with 1× HBSS (Gibco) containing 15 mM HEPES (Thermo Fisher Scientific). Peyer´s patches were carefully excised from SI. SI and LI were opened longitudinally, cut into approx. 1 cm pieces and washed three times with 1× HBSS (Gibco) containing 15 mM HEPES (Thermo Fisher Scientific), 5% FCS (Viralex™, PAA Laboratories), 50 µg/mL gentamycin (Gibco), and 2 mM EDTA (Invitrogen, Life Technologies)

for 15 min at 37 °C. During the first incubation step, 0.15 mg/mL DL-dithiothreitol (Sigma–Aldrich) was added to LI samples. LI, but not SI samples, were shaken on an orbital shaker at 350 rpm during incubation. After each incubation step, SI, but not LI samples, were shaken by hand for 10 s. Media containing epithelial cells and debris were discarded by filtration through a 250 µm mesh (Tekniska Precisionsfilter, JR AB). The remaining tissue was digested for 20–28 min at 37 °C under magnetic stirring (350-500 rpm) in R-10 media (RPMI 1640 (Gibco) with 10% FCS (Viralex, PAA Laboratories), 10 mM HEPES (Thermo Fisher Scientific), 100 U/mL penicillin (Gibco), 100 µg/mL streptomycin (Gibco), 50 µg/mL gentamycin (Gibco), 50 µM 2-mercaptoethanol (Gibco), and 1 mM sodium pyruvate (Gibco)) containing 1 mg/ml collagenase P (Roche) and 0.03 mg/mL DNase I (Roche). After digestion, SI-LP and LI-LP cells were purified by density gradient centrifugation (600 rcf for 20 min at 22 °C, acceleration 5 and brake 0) with 40/70 % Percoll (GE Healthcare). Cell suspensions were subsequently filtered through 100 µm cell strainers (BD Biosciences) and restimulated in vitro prior to staining for flow cytometric analysis.

**Isolation of liver cells**. To isolate liver cells, livers were collected from 1× PBS (Gibco)-perfused mice, cut into small pieces and digested for 40 min at 37 °C in R-10 media (RPMI 1640 (Gibco) with 10% FCS (Viralex, PAA Laboratories), 10 mM HEPES (Thermo Fisher Scientific), 100 U/mL penicillin (Gibco), 100 µg/mL streptomycin (Gibco), 50 µg/mL gentamycin (Gibco), 50 µM 2-mercaptoethanol (Gibco), and 1 mM sodium pyruvate (Gibco)) containing 0.3 mg/ml collagenase IV (Roche), 0.21 mg/mL collagenase D (Roche) and 0.05 mg/mL DNase I (Roche) on an orbital shaker (370 rpm). After digestion, liver cells were filtered through 100 µm cell strainers (BD Biosciences) and subsequently purified by density gradient centrifugation (800 rcf for 20 min at 22 °C, acceleration 5 and brake 0) with 40/70% Percoll (GE Healthcare). Cell suspensions were either restimulated in vitro prior to or directly used for flow cytometric analysis.

**Ex vivo stimulation of SI-LP, LI-LP, and liver cells**. SI-LP, LI-LP, and liver cells were restimulated in vitro in R-10 media in the presence of either 20 ng/mL IL-23 (R&D Systems) or 250 ng/mL PMA (Sigma–Aldrich) in combination with 0.5 µg/mL ionomycin (Sigma–Aldrich) for 4 h at 37 °C. After 1 h incubation, 10 µg/mL brefeldin A (BioLegend) was added.

**Flow cytometry**. Flow cytometry was performed according to standard procedures. Cell aggregates (identified in FSC-H or FSC-W vs. FSC-A scatter plots) and dead cells identified by using Zombie Aqua or Zombie UV Fixable Viability Kit (BioLegend) were excluded from analyses. Intracellular staining was performed using the eBioscience FoxP3/Transcription Factor Staining Buffer Set (eBioscience) according to the manufacturer's instructions. Data was acquired on a LSRII (BD Biosciences) and analyzed using FlowJo software (Tree Star).

**Antibodies (Abs) and reagents**. The following mAbs and reagents were used in the study: anti-CD3ε (17A2), anti-CD4 (GK1.5), anti-CD8α (53-6.7), anti- CD11b (M1/70), anti-CD11c (N418), anti-CD19 (6D5 and ID3), anti-CD45 (30F11), anti-CD45R/B220 (RA3-6B2), anti-CD90.2 (30-H12), anti-CD127 (A7R34), anti-Ly-6G/Ly6C (Gr-1) (RB6-8C5), anti-Ly6C (AL-21), anti-Ly-6G (1A8), anti-TER-119 (TER-119), anti-F4/80 (BM8), anti-I-A/I-E (M5/114.15.2), anti-Siglec-F (E50-2440), anti-TIM-4 (54(RMT4-54)), anti-TCRβ (H57-597), anti-TCRγδ (GL3), anti-NK1.1 (PK136), anti-NKp46 (29A1.4), anti-FoxP3 (FJK-16s), anti-RORγt (B2D), anti-T-bet (4B10), anti-Ki67 (B56), anti-IL-10 (JES5-16E3), anti-IL-17A (TC11-18H10.1), anti-IL-22 (IL22JOP), and anti-IFNγ (XMG1.2). All Abs were purchased from eBioscience, BioLegend or BD Biosciences. PECF594-conjugated and BV421-conjugated streptavidin were purchased from BD Biosciences and BioLegend, respectively.

**Multiplex cytokine quantification**. Liver tissues were crushed with a mortar and pestle on liquid nitrogen. 30–40 mg of tissue were transferred to a clean tube and 400-µL T-PER tissue protein extraction reagent (Thermo Scientific) buffer including proteinase inhibitors (Sigma–Aldrich) were added. Protein concentration was assessed by BCA (Thermo Scientific) according to manufacturer's instruction. Cytokines were quantified using xMAP technology with combinations of Bio-Plex Pro (BioRad) premixed mouse cytokine panels, and the analyses were carried out on a Bio-Plex 200 system and the Bio-Plex Manager Software package.

**Statistical analysis**. Omics data were acquired as described in their appropriate sections. Normality residuals of remaining data were assessed by D'Augostino-Pearson omnibus (k2), Graphpad Prism 8, and subsequently evaluated by parametric (gaussian distributed) or non-parametric (non-gaussian distributed) tests as appropriate. Details of each test including post-hoc assessment are specified in figure legends. Data are expressed as indicated in individual figure legends, and all groups are compared by two-sided tests to the relevant WD$_{REF}$ or WD$_{CNTL}$ group as indicated.

**Reporting summary**. Further information on research design is available in the Nature Research Reporting Summary linked to this article.

## Data availability

All data are in this manuscript are available on reasonable request to the corresponding authors. 16S sequencing data are uploaded to a public repository with study accession number: PRJEB41983 and available at the following link: http://www.ebi.ac.uk/ena/data/view/PRJEB41983. All non-omics datasets included in the current study are available in the figshare repository Jensen, B. A. H. et al. Data underlying the manuscript: Lysates of Methylococcus capsulatus Bath induce a lean-like microbiota, intestinal FoxP3+RORγt + IL-17+ Tregs and improve metabolism. figshare https://doi.org/10.6084/m9.figshare.13522283 (2021).

## Code availability

Standard codes from the cited R-packages were used in this manuscript. Specific scripts are available upon reasonable request.

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

## Acknowledgements

This work was supported by the Norwegian Research Council (project 267655). B.A.H.J. was supported by Lundbeck Foundation (grant number: R232-2016-2425) and Novo Nordisk Foundation (grant number: NNF17OC0026698). A.-L.G. and A.M. were supported by Canadian Institutes for Heart Research and Sentinel North from the Canada First Research Excellence Fund.

## Author contributions

B.A.H.J., J.B.H., C.R.K., K.K., and T.E.L. conceived and designed the study. B.A.H.J., J.B.H., I.S.L., S.B.S., and M.T.F.D. conducted the in vivo experiments. B.A.H.J., K.K., and T.E.L. supervised all parts of the study. J.T.T., A.M., L.M., W.A., and C.S. supervised parts of the study. N.v.B., S.I.P., and A.R. performed flow cytometry analyses. B.A.H.J., J.B.H., I.S.L., S.D., C.P.Å., N.v.B., A.L.A., A.R., S.A.I., Y.J.A., K.S., E.F., M.T.F.D., and SIP performed key experiments and analyzed the data. B.A.H.J. integrated the data and wrote the manuscript with inputs from J.B.H., I.S.L., K.K., and T.E.L. All authors edited, revised, and approved the final version of the manuscript.

## Competing interests

B.A.H.J., J.B.H., I.S.L., K.K., C.R.K., and T.E.L. are co-inventors of International (PCT) Patent Application No. PCT/EP2018/071076 based on the enclosed data. The remaining authors declare no competing interests.
