## [Peer Review File · Nature Communications]

Reviewers' comments:

Reviewer #1, expert in immunometabolism (Remarks to the Author):

In the paper by Jensen and colleagues, the authors analyzed the role of *Methylococcus capsulatus* Bath on the metabolic alteration induced by western diet. More specifically, they found that intake of whole-cell lysates of the non-commensal bacterium *Methylococcus capsulatus* Bath (McB) was able to reverse high fat/high sucrose-induced changes in the gut microbiota to a state similar of that observed in lean mice. These events were associated with induction of Foxp3+ROR γ t+IL-17+ regulatory T cells in the small intestine and colon, and enhanced mucus production, suggesting improved gut health. Moreover, mice receiving diet supplemented with McB lysates exhibited improved glucose regulation, reduced body and liver fat along with diminished hepatic immune infiltration, which protected them from NAFLD development.

The topic of the manuscript is of potential interest, as it suggests McB lysate as a potent modulator of immunometabolic homeostasis, being able to change gut microbiota. Despite the novelty of the manuscript, in the present form, this referee feels that there are several issues which limit the enthusiasm for the manuscript and that should at least better addressed to further sustain the conclusion presented in this work.

Major concerns:

1) In Figure 2, the authors showed that WD MCB (*Methylococcus capsulatus* Bath) feeding was able to stimulate induction of gut-specific regulatory T cells. However, only their percentage (gated on CD4) and the expression of IL-17 and ROR γ t within this subset were analyzed, while rather also their absolute number should be provided.

Moreover and more importantly, no functional assays on purified Treg cells from the different mice (fed with different diets) have been performed. The authors should better define whether WD MCB feeding is able also to alter Treg cell suppressive activity, their own proliferation, or cytokine production of regulatory/anti-inflammatory cytokines, thus contributing to the inhibition of the inflammatory process.

The authors are strongly recommended to add this info.

2) To define the direct role of *Methylococcus capsulatus* Bath on the modulation of immune response, the authors should perform a series of experiments also *in vitro*. For example, the authors should demonstrate whether dendritic cells pulsed with McB are able to differently modulate T cell function, or whether they have any effect on the process of Treg cell induction/generation.

These experiments would allow us to understand if the effect of McB on the immune system is direct (thus explaining the *in vivo* data) or if it is mediated by changes in the microbiota.

3) In figure 5, the authors showed that WD MCB feeding was able to reset the hepatic lipidome and decrease hepatic immune infiltration, alleviating NAFLD. In this regard, they only showed CD3+ and Ly6G+ (neutrophils) cells in the liver of treated mice. It would also be appropriate providing information on the Treg infiltrating the liver, given the role previously described of *Methylococcus capsulatus* Bath on colonic Treg cells. In parallel, a more detailed analysis of hepatic lymphocyte activation should be performed (ie. by evaluating the expression of activation markers), in order to understand whether the beneficial effect of McB is to be ascribed to a reduced hepatic immune recruitment associated with an altered activation of immune cells.

Minor concerns:

1) The authors in figure 4 evaluated the effects of WD MCB feeding on NAFLD activity score, lipid droplets, lipid size and adiponectin levels in liver samples of treated mice. In this context, also plasma leptin and resistin levels together with cholesterol (LDL and HDL) and triglycerides levels in McB fed mice should be evaluated.

Reviewer #2, expert in metabolic impact of microbiota (Remarks to the Author):

This is a comprehensive set of data that explores the effects of McB on colon immune function, metabolic factors, microbiota and metabolites, and liver outcomes when given in conjunction with a western diet. The McB effects appear promising.

Were there any attempts to examine mechanisms eliciting cross talk between microbiota and metabolism?

It is difficult to decipher whether it is the reduction in body weight that is causing the effects on immune function, metabolism etc. or whether it is a direct effect of McB. A weight-matched group should have been included as an appropriate control. This is a major flaw in the study design. If mice were weight matched we would be able to determine if all of the changes occurring with the bacteria were direct effects or a result of the changes in weight.

A second major weakness of the study is the lack of exact mechanisms that may explain the benefits of McB on metabolic improvement. While this reviewer appreciates the comprehensive set of data that the authors generated, it is disappointing that mechanisms were not explored.

Point-by-Point response MS_NCOMMS-19-37264

Reviewers' comments:

Reviewer #1, expert in immunometabolism (Remarks to the Author):

In the paper by Jensen and colleagues, the authors analyzed the role of *Methylococcus capsulatus* Bath on the metabolic alteration induced by western diet. More specifically, they found that intake of whole-cell lysates of the non-commensal bacterium *Methylococcus capsulatus* Bath (McB) was able to reverse high fat/high sucrose-induced changes in the gut microbiota to a state similar of that observed in lean mice. These events were associated with induction of Foxp3⁺RORγt⁺IL-17⁺ regulatory T cells in the small intestine and colon, and enhanced mucus production, suggesting improved gut health. Moreover, mice receiving diet supplemented with McB lysates exhibited improved glucose regulation, reduced body and liver fat along with diminished hepatic immune infiltration, which protected them from NAFLD development.

The topic of the manuscript is of potential interest, as it suggests McB lysate as a potent modulator of immunometabolic homeostasis, being able to change gut microbiota. Despite the novelty of the manuscript, in the present form, this referee feels that there are several issues which limit the enthusiasm for the manuscript and that should at least better addressed to further sustain the conclusion presented in this work.

Major concerns:

1) In Figure 2, the authors showed that WD MCB (*Methylococcus capsulatus* Bath) feeding was able to stimulate induction of gut-specific regulatory T cells. However, only their percentage (gated on CD4) and the expression of IL-17 and RORγt within this subset were analyzed, while rather also their absolute number should be provided.

Response: We appreciate the input and recognize that in certain situations absolute numbers may be more informative. We would, however, also like to emphasize the risk of absolute cell numbers to artificially inflate certain subpopulations, where differences might solely be driven by specific traits in the parent population. This is most notably observed in our newly included data on IL-10 and Ki67 expression. Here, we observed a vastly increased amount of Ki67⁺ pT_{regs} in WD_{McB}-fed mice. At face value, increased numbers of Ki67⁺ cells in WD_{McB}-fed mice point towards enhanced proliferation. Still, as evidenced by the relative percentages of Ki67⁺ pT_{regs}, this is not true. Specifically, no differences were observed between groups in the percentage of Ki67⁺ n/pT_{regs}, indicating that the apparent increase of Ki67⁺ pT_{regs} in WD_{McB}-fed mice solely reflected the absolute numbers of the parent cell type (i.e. pT_{regs}). Thus, the reason behind this increased cell number, was not enhanced proliferation capacity, but rather either recruitment or *in situ* differentiation. The latter has been suggested to be facilitated by bacterial metabolism of bile acids (Campbell *et al.*, 2020; Song *et al.*, 2020) as well as SCFAs in general (Smith *et al.*, 2013) and butyrate in particular (Furusawa *et al.*, 2013). In part, the same phenomenon applies in relation to IL-10 expression. Here we observed increased amounts of IL-10⁺ pT_{regs} in both SI- and LI-LP, corroborating advanced

immunosuppression in both segments. Yet, it was only in LI-LP that the relative amounts of $IL10^+ pT_{regs}$ (and nT_{regs} for that matter) were increased, suggesting that the phenotypic shift of enhanced secretory capacity was restricted to the colon of WD_{Mcb} -fed mice.

Moreover, as stated in the original MS and also noted in Figure 1 and 2, we compiled the data from two independent experiments (visually separated by experiment-specific dot shapes). Considering pronounced experiment-to-experiment variation of live Lamina Propria (LP) extracted cells, (as also clearly presented in Figure 2A and 2H, showing the absolute numbers of viable $CD4^+$ T cells), we strongly believe that depicting the relative proportion of each cell subset is a more relevant approach, as absolute numbers are confounded by immense inter-experiment variation, which is particularly important for the comparison between WD_{REF} - and LFD-fed mice. As the latter was only included in one of the two experiments, any variation might unjustifiably compromise statistical significance in a situation of pronounced biological significance (comparing the circled dots of those two groups while omitting the squared dots only present in the WD_{REF} -fed group). This would also better align with the depicted dot-plots from flow cytometry (Figure 2E-G, L-N), hence allowing the reader to evaluate whether we rightfully have chosen group medians as representative, or if we had manipulated the representation by showing e.g. the best responding WD_{Mcb} -fed mouse compared to its lowest/least responding WD_{REF} -fed counterpart. We would thus like to keep the relative percentages of these subgroups in the main figures. Still, respecting the comment of this reviewer, we now provide absolute numbers of the observed Treg phenotype for all relevant subtypes in new supplementary figures (S3F-G; S4C-F).

Action: We now report absolute numbers of gut-specific regulatory T cells (Figure S3F) and their phenotypic subsets ($IL17^+$, $IL10^+$, $Ki67^+$, S3G; S4C-F) to meet this reviewer's request.

Moreover and more importantly, no functional assays on purified Treg cells from the different mice (fed with different diets) have been performed. The authors should better define whether WD_{Mcb} feeding is able also to alter Treg cell suppressive activity, their own proliferation, or cytokine production of regulatory/anti-inflammatory cytokines, thus contributing to the inhibition of the inflammatory process.

The authors are strongly recommended to add this info.

Response: We appreciate the suggestions of functional assays, but would also like to emphasize the lack of feasibility for *ex vivo* immunosuppressive assays using primary pT_{reg} cells isolated from SI- and LI-LP. In short, Treg populations would need to be sorted out from SI- and LI-LP prior to subsequent functional assays. Such experimental strategy is technical impossible, in part due to limited cell numbers of this subpopulation.

In light of this reflection, and to still meet the request of this reviewer, we designed new experiments in weight matched mice, determining IL-10 production and Ki67 expression, a hallmark immunosuppressive cytokine instrumental for the suppressive capacity of Tregs and a proliferation marker, respectively.

These new results confirmed our original claims of an WD_{Mcb} -induced immune suppression, corroborated by increased numbers of $IL10^+ pT_{regs}$ in both SI- and LI-LP, and further points towards colon-specific WD_{Mcb} -induced shift in pT_{reg} phenotype, as only pT_{regs} (and nT_{regs}) at this site increased their secretory capacity of IL-10 (indicated by the *relative* proportion of $IL-10^+ n/pT_{regs}$).

While enhanced IL-10 production was indeed observed in both p^- and nT_{regs} from LI-LP of WD_{McB} -fed mice, $IL-10^+ pT_{regs}$ abundantly outnumbered their thymus derived counterparts (nT_{regs}), hence supporting the proposed involvement of gut specific pT_{reg} in McB-induced immune suppression. Lastly, although absolute numbers of $Ki67^+ pT_{regs}$ were increased in WD_{McB} -fed mice, this was simply a reflection of the abundantly increased pT_{reg} number, as no difference in the relative amount of $Ki67^+ n/pT_{regs}$ was observed between groups.

Action: We designed new experiments in weight matched mice, determining IL-10 production and Ki67 expression in naturally occurring (n^-) and gut specific (p^-) T_{regs} in both SI- and LI-LP, included in new Figure 2O-R, S4C-F.

2) To define the direct role of *Methylococcus capsulatus* Bath on the modulation of immune response, the authors should perform a series of experiments also *in vitro*. For example, the authors should demonstrate whether dendritic cells pulsed with McB are able to differently modulate T cell function, or whether they have any effect on the process of Treg cell induction/generation.

These experiments would allow us to understand if the effect of McB on the immune system is direct (thus explaining the *in vivo* data) or if it is mediated by changes in the microbiota.

Response: While we appreciate the suggestion, the requested data has already been published (Indrelid *et al.*, 2017). Intriguingly, this report demonstrate a prominent interaction of our McB lysate with human DCs affecting T cell responses in co-culture experiments. Specifically, when compared to *E. coli* Nissle 1917 (EcN) and LGG experiments, McB lysates induced an *IL-2-driven* DC response. Out of the 8 cytokines measured, of which EcN affected all and LGG none, McB exclusively affected secretion of IL-2 (in notable amounts), a hallmark cytokine for T cell activation inducing proliferation of both effector and regulatory T cells. Enhanced T cell proliferation was corroborated by 50% increased CD25 expression (IL-2 alpha receptor) and augmented 3H thymidine incorporation (cell division marker). Unfortunately, ROR γ t expression was not measured in this report (Indrelid *et al.*, 2017), and so we do not know the extent to which McB lysate stimulate gut-specific Treg induction *in vitro*.

To meet the request of this reviewer, we have performed RNAseq data on DCs stimulated with McB lysates and find a specific upregulation of galactin-10, which has been shown to be instrumental for the suppressive capacity of naturally occurring Tregs (Kubach *et al.*, 2007), thus strongly supporting the hypothesis of a direct McB instruction, on at least human DCs, to affect Tregs polarization and/or function.

Nevertheless, we refrained from including these data to avoid overinterpretation in this preliminary form and instead submit them for the reviewers' eyes only (Table R1). We envisage that such data would be better suited for a paper targeting solely the immunomodulatory actions of McB aiming at identifying biologically relevant effector molecules, once their presence *in situ* has been confirmed. We sincerely hope this reviewer supports our decision to improve transparency and reproducibility.

Action: We refer to the previously published work in the revised introduction (line 117-118) and discussion (line 722-723).

3) In figure 5, the authors showed that WD McB feeding was able to reset the hepatic lipidome and decrease hepatic immune infiltration, alleviating NAFLD. In this regard, they only showed CD3+

and Ly6G⁺ (neutrophils) cells in the liver of treated mice. It would also be appropriate providing information on the Treg infiltrating the liver, given the role previously described of *Methylococcus capsulatus* Bath on colonic Treg cells. In parallel, a more detailed analysis of hepatic lymphocyte activation should be performed (i.e. by evaluating the expression of activation markers), in order to understand whether the beneficial effect of McB is to be ascribed to a reduced hepatic immune recruitment associated with an altered activation of immune cells.

Response: We appreciate the suggestion and absolutely agree that such data would be helpful to better characterize potential compartmentalization of McB-induced immune responses. We therefore initiated new *in vivo* experiments focusing on the hepatic immune profile in weight matched mice. Surprisingly, this experiment revealed increased numbers (both the relative fraction of CD45⁺ cells and in absolute numbers) of tissue resident Tim4⁺ macrophages (i.e. Kupffer cells) in WD_{McB}-fed mice, suggesting that McB feeding either recruits or stimulate *in situ* proliferation of this key cell subset driving hepatic homeostasis. Kupffer cells are central to innate immunity and responsible for containment and clearance of foreign particles. Inflammatory activation of hepatic Kupffer cells potentiates obesity-associated insulin resistance, in part by recruiting neutrophils and T cells (Baffy, 2009). Still, Kupffer cells exhibit tremendous plasticity in their activation program, with anti-inflammatory properties in their alternative activation state ameliorating hepatic steatosis (Odegaard *et al.*, 2008). While our staining panel did not allow us to identify the activation state of the enhanced Kupffer cell proportions, it is pertinent to note that none of the classically recruited cell types were altered in numbers (Figure S5J,K). Instead, we observed increased proportion of Ly6C⁺ monocytes. Interestingly, the mean fluorescence intensity (MFI) within those monocytes in our new short term experiment appeared lower in McB-fed mice than in their WD_{REF}-fed counterparts (Figure 5J). Newly recruited monocytes express high levels of Ly6C in their inflammatory state; an expression that is gradually downregulated in immune resolving alternatively activated cells (Ramachandran *et al.*, 2012; Dal-Secco *et al.*, 2015; Ju and Tacke, 2016).

Additionally, and despite weight maintenance, WD_{REF}-fed mice exhibited ~50% increase in hepatic IL-17⁺ $\gamma\delta$ T-cells (Figure 5K), an immunological precursor for subsequent NAFLD controlled by the gut microbiota (Li *et al.*, 2017). WD_{McB}-fed mice were fully protected from this trait, pointing towards extraintestinal regulation of innate immunity key to metabolic homeostasis.

Pointing towards a compartmentalized immune-modulation, we did not observe any differences in the amount of intestinal IL-17⁺ $\gamma\delta$ T-cells nor hepatic n/p T_{regs}. It should be noted, though, that ROR γ t⁺, FoxP3⁺ Tregs are known to exclusively reside in the gastrointestinal immune system, and thus, the low frequencies observed in the hepatic profile (<1% contrasting the 3-12% and 20-30% in SI- and LI-LP, respectively, Figure S4B) is not surprising.

Action: We have included the above-described data in Figure 5I-K; S4A-B; S5I-K, extended the results section (**line 452-452; 548-566**) and amended the discussion accordingly (**line 731-739**).

Minor concerns:

1) The authors in figure 4 evaluated the effects of WD McB feeding on NAFLD activity score, lipid

Point-by-Point response MS_NCOMMS-19-37264

droplets, lipid size and adiponectin levels in liver samples of treated mice. In this context, also plasma leptin and resistin levels together with cholesterol (LDL and HDL) and triglycerides levels in McB fed mice should be evaluated.

Response: We agree that many of these suggestion would be of interest (e.g. plasma resistin and triglycerides (TG)), but unfortunately we do not have material left to analyze these endpoints in obese mice. The amount of body fluids, including plasma, is extremely limited in mice, hence the shortage. Instead, we had small amounts of plasma left from a few mice in the original 12+6 weeks experiment (first presented), from which we observed numerically decreased TG and cholesterol levels in WD_{McB}-fed mice, corroborating our histologically assessed data from the prolonged and intensified 21+5 week setup (please see Figure R1). Unfortunately, we did not have enough sample to assess resistin, leptin, and HDL:LDL cholesterol.

We would, however, like to point the attention to line 512-513 where we write the following: “... further supported by the assessment of insulin tolerance and hepatic gene transcription activity of genes encoding key metabolic enzymes (Figure 4I-J)...”. In our opinion, these data strongly corroborate diminished insulin resistance in WD_{McB}-fed mice. Thus, while resistin and leptin are relevant markers of insulin resistance, potentially complementing our adiponectin data, we did report improved insulin sensitivity as assessed by an insulin tolerance test, and so additional markers are likely to simply follow a similar pattern to the already reported data.

Lastly, while certain genotypes of mice have been developed as suitable tools for assessing the cholesterol profile (e.g. LDL^{r/-}ApoB^{100/100} and ApoE^{-/-}), WT mice – as used here – are well-known to have a tremendously skewed, none-human-like, HDL to LDL profile, and so we respectfully disagree that the LDL to HDL ratio would be informative in this setting. Regardless of its suitability, it was not possible for us to evaluate the requested profile.

Although the newly generated data support and corroborate our original claims, we do not feel confident to include these data, considering the low n-number combined with extended shelf-life. We therefore request (also considering the amount of data already included) that we keep these data for the reviewer’s eyes only. This will further enable us to focus on this concept in more suitable experimental settings in the future (mouse genotype, n-number, reproducibility, etc.).

Action: We have analyzed TG and cholesterol profiles in subsets of mice included in Figure R1.

Reviewer #2, expert in metabolic impact of microbiota (Remarks to the Author):

This is a comprehensive set of data that explores the effects of McB on colon immune function, metabolic factors, microbiota and metabolites, and liver outcomes when given in conjunction with a western diet. The McB effects appear promising.

Response: We appreciate the positive evaluation of the substantial amount of work included in this manuscript and hope that the revised version will spur further enthusiasm.

Were there any attempts to examine mechanisms eliciting cross talk between microbiota and metabolism?

Response: We have, in the revised version, conducted a series of *in vivo* experiments to enhance our understanding of the mechanisms whereby McB improves immunometabolism. To specifically

address if the observed gut microbial normalization following McB feeding improved metabolism rather than representing a simple bystander effect, we transferred cecal microbiota from WD_{REF}-, WD_{McB}- and LFD-fed donor mice to either LFD- or WD-fed recipient mice and assessed weight and fat mass development as well as their glucoregulatory capacity. While cecal microbiota transfer (CMT) did not affect any of the measured outcomes in LFD fed mice, WD-fed recipient mice generally mirrored the phenotype of their respective donors despite not reaching statistical significance, suggesting that the cecal microbiota at least partially influenced the metabolic phenotype. Future studies with increased power is warranted to elucidate the full influence of the observed microbial adjustments.

Action: We transferred cecal content from WD_{REF}-, WD_{McB}- and LFD-fed donor mice to either LFD- or WD-fed recipient mice and assessed weight and fat mass development as well as their glucoregulatory capacity as presented in new Figure 8 and Figure S6. The results sections and discussion are amended accordingly.

It is difficult to decipher whether it is the reduction in body weight that is causing the effects on immune function, metabolism etc. or whether it is a direct effect of McB. A weight-matched group should have been included as an appropriate control. This is a major flaw in the study design. If mice were weight matched we would be able to determine if all of the changes occurring with the bacteria were direct effects or a result of the changes in weight.

Response: We recognize the inherent challenge of weight differences in mouse experiments potentially confounding data interpretation. There is no easy fix to this challenge, as weight matched groups of mice receiving a more obesogenic diet, would have to be calorie restricted (or subjected to exercise) to remain lean, both of which profoundly affect host metabolism. We would also like to emphasize that while we observed significant body weight differences in the 12+6 week protocol, primarily caused by weight maintenance of WD_{McB} feeding and thus not a weight reduction *per se* (please consult Figure 3H-I), we did not observe notable differences in the intensified 21+5 weeks protocol (Figure S5B-C), where WD_{REF}-fed mice exhibited plateaued weight development in this prolonged setup. Despite this weight (but not fat) dissimilarity between the experiments, we still observed improved insulin sensitivity and gut immunity along with a ‘normalized’ gut microbiota, strongly pointing towards a direct impact of WD_{McB} feeding on host immunometabolism.

Still, to evaluate the consistency of this trait we first subdivided mice from the 12+6 weeks protocol into groups of similar final bodyweight and evaluated their relative protection against impaired insulin sensitivity (Figure R2). The main challenge with this setup is, however, that while final body weight is similar (obtained by including the most obese WD_{McB}-fed mice and compared to their leanest WD_{REF}-fed counterparts), their 12+0 weeks baseline weight and glucose handling diverged considerably. This discrepancy roots in mouse to mouse variation as all mice at this time point were fed identical diets. Yet, despite these discrepancies, WD_{McB}-fed mice were still protected against weight gain and enhanced insulin secretion during a glucose challenge, observed in their WD_{REF}-fed counterparts (Figure R2). Considering the inconsistent baseline results with this post hoc test, we refrain from including these data in the revised MS and instead provide them for the reviewers’ eyes only.

Yet, to improve our understanding of the molecular impact by WD_{McB}-feeding in weight matched mice, we next designed a short term study and euthanized the mice for gut and hepatic immune assessment before the weight significantly diverged (Figure S5H). These results corroborate a highly reproducible impact on gut immunity, even in the absence of weight differences. As such, WD_{McB}-fed mice presented a unique phenotype with superior pT_{reg} induction along with enhanced IL-10 secretory capacity compared to both LFD- and WD_{REF}-fed mice; a trait that persisted in both intestinal segments in relation to pT_{reg} induction as well as the amount of IL-10⁺ positive cells (Figure S4B-D). Notably, enhanced secretor capacity was restricted to LI-LP (Figure 2O-P). Moreover, WD_{McB}-fed mice were also protected from the WD_{REF}-induced blooms in hepatic IL-17⁺ γδ T cells abundances (Figure 5K). IL-17⁺ γδ T cells both precede and induce NAFLD, and are generally controlled by the gut microbiota (Li *et al.*, 2017), thus pointing towards an McB-controlled modulation of the WD-affected gut-liver axis.

Action: We provide the requested data on the glucoregulatory phenotype in post hoc analyses of weight-stratified mice from the 12+6 week experiment as Figure R2, and have further included a substantial amount of new data from short term experiments in ‘weight-matched’ mice (Figure 2O-R; 5I-K; S1C; S4A-F). Both results section and discussion are amended accordingly.

A second major weakness of the study is the lack of exact mechanisms that may explain the benefits of McB on metabolic improvement. While this reviewer appreciates the comprehensive set of data that the authors generated, it is disappointing that mechanisms were not explored.

Response: We appreciate this reviewer’s enthusiasm for the comprehensive set of data provided, but obviously regret that we have disappointed in terms of identifying causal relations. Such data were originally planned for follow-up articles both considering the amount of data already provided and not least the complex relationship between host immunity, microbe normalization and improved metabolism. It was thus a deliberate decision not to chase down incomprehensive causal links potentially oversimplifying biological relevance and the hierarchy between identified traits. That being said, we also realize the importance of causality when submitting to top-tier journals, and similarly admit that the originally submitted MS inadequately addressed such concerns. We have therefore conducted a series of experiments, as also outlined above (and illustrated in new Figure S1C,D), to first pinpoint the possible importance of adaptive immunity in the McB-induced gut microbial and metabolic ‘normalization’. We further assessed if gut microbial alterations induced by McB feeding affected host metabolism and thirdly the extent to which McB lysates and adaptive immunity synergized to control gut microbial composition.

We identified especially a single symbiont (namely *Parabacteroides*), where both the ~10-fold augmented abundance induced by WD_{McB} feeding and the ~3-fold diminished abundance induced by WD_{REF} feeding were dependent on adaptive immunity, contrasting the remarkably stable abundances observed in LFD-fed reference mice (Figure 7A-C,E; S6A-B). The WD_{McB}-induced change in this genus was a major reason for the observed shift in the microbiome consortium in WD_{McB}-fed WT mice (Figure 7D,G). This general change in community structure, did not precipitate in WD_{McB}-fed RAG2^{-/-}, where *Parabacteroides* abundances were similarly unaffected by experimental diets (Figure 7E,H).

It is worth noting, that not only did the prevalence of *Parabacteroides* not differ between all groups and time points in RAG2^{-/-} fed mice, the relative abundance of this genus in these mice was also lowered to a level resembling that of WD_{REF}-fed WT mice (thus ~3-fold lower than LFD-fed WT mice, Figure 7A-C,E; S6A), hence pointing towards a strong involvement of adaptive immunity to support an ecological niche especially favourable for *Parabacteroides*. Lastly, these data indicate that WD_{McB} feeding synergizes with adaptive immunity to facilitate *Parabacteroides* growth (Figure 7A-C,E).

Other commensals, such as *Barnesiella*, *Allobaculum*, *Clostridium IV*, and *Desulfovibrio* remained, as *Parabacteroides*, stable over time in LFD-fed mice, but were substantially and inversely regulated between WD_{REF}- and WD_{McB}-fed mice (Figure S6A,B). The feed and time dependent trajectories of these specific bacteria were independent of adaptive immunity and thus similarly regulated in both WT and RAG2^{-/-} mice.

We next evaluated the glucoregulatory capacity in RAG2^{-/-} mice fed either diet for 6 weeks and observed a partial protection against WD-induced 5h fasted hyperinsulinemia, insulin secretion during OGTT, body weight gain, and total fat mass (Figure 8A-G). Both cecum weight and -SCFAs were further increased to levels resembling those in the above-reported WT mice (Figure 8H,I). Collectively, these data suggest that at least some of the metabolic effects observed in WD_{McB}-fed mice may occur independent of adaptive immunity.

We therefore designed cecal microbiota transfer (CMT) experiments in ABX treated mice and assessed the glucoregulatory capacity in cohorts fed either LFD or WD_{REF} (Figure S1D). While we failed to observe a donor-dependent effect of CMT in LFD-fed mice (Figure 8J,K; S6D-G), we did observe transient tendencies of metabolic improvements in WD-fed mice receiving cecal microbes from LFD- or WD_{McB}-fed donors (Figure 8K-O).

Action: We have conducted 1) new experiments in RAG2^{-/-} mice to elucidate the impact of adaptive immunity on WD_{McB}-induced metabolic traits, 2) CMT experiments of LFD-, WD_{REF}-, and WD_{McB}-fed donor mice to both LFD- and WD-fed recipients (n = 9-12 per recipient-group per diet) to investigate the importance of WD_{McB}-instructed gut microbial alterations in the onset of diet induced obesity, and 3) assessed the extent to which McB lysates and adaptive immunity synergize to control gut microbial composition. The result sections (primarily Figures 7, 8 & S6) and discussion are amended accordingly.

While further studies are warranted to fully disentangle the complex relationship between the induced traits, we hope that the revised manuscript provides sufficient indications of potential causal links to warrant publication in NCOMMS.

Table R1: Galectins differentially expressed in DCs pulsed with McB, *Escherichia coli* Nissle 1917 (EcN) and *Lactobacillus rhamnosus* GG (LGG).

Point-by-Point response MS_NCOMMS-19-37264

Gene symbol	Gene name	Log(fold change)	Log(fold change)	Log(fold change)	Reported roles in determining T cell fate
		LGG vs A	EcN vs A	McB vs A	
LGALS1	Galectin-1		-1,2024		Galectin 1 activation induces increased levels of IL-27 and IL-10 production, suppresses Th1/Th17 cell differentiation and promotes T-regulatory responses (Ilarregui et al., 2009; Toscano et al., 2007).
LGALS2	Galectin-2	-2,46125	-2,15071		Induces apoptosis of lamina propria T cells in vitro (Sturm et al., 2004).
LGALS3	Galectin-3	2,25384			Suppresses Th17 responses by regulating DC cytokine production and regulates Th1/Th2 balance by affecting IL-12 production (Fermin Lee et al., 2013; Bernardes et al., 2006)
LGALS8	Galectin-8	1,7791			Galectin-8 promotes regulatory T cell differentiation by modulating IL-2 and TGF β signaling (Sampson, Suryawanshi, Chen, Rabinovich, & Panjwani, 2016).
LGALS9	Galectin-9	0,951287			DC galectin-9 interactions with Tim-3 on activated T cells induces Th1 cell apoptosis promote T-regulatory cell differentiation and suppresses Th17 cell differentiation (Zhu et al., 2005; Sehwat, Suryawanshi, Hirashima, & Rouse, 2009; Seki et al., 2008).
CLC	Charcot-Leyden crystal, galectin 10			2,61782	Critical for the suppressive function of CD4+CD25+ Tregs and functions as a T cell-suppressive molecule in regulatory eosinophils (Kubach et al., 2007; Lingblom, Andersson, Andersson, & Wenneras, 2017).

Methods for Table R1:

Bacterial stimulation, RNA extraction, amplification and labeling

MoDCs were primed for 24 h by UV-inactivated bacteria in a ratio of 1:100 (MoDC: bacteria) or with a maturation cocktail of 15 ng/ml TNF- α (ImmunoTools), 100 ng/ml LPS and 5 μ g/ml PGE2 (Sigma-Aldrich). Cells were harvested and RNA isolated by Maxwell RSC simplyRNA Cells Kit using the automated Maxwell RSystem. Quantification was performed using a spectrophotometer (NanoDrop Technologies), and RNA quality was assessed by Agilent Bioanalyzer 2100. All RIN values were above 8.4. RNA was prepared for sequencing using the Strand-specific TruSeqTM RNA-seq kit and single read (unpaired) 75-basepair sequencing was performed using the HiSeq platform (Illumina Inc., San Diego, California, USA) at The Norwegian Sequencing Centre.

Bacterial strains and culture conditions

M. capsulatus (Bath) NCIMB11132 (GenBank accession number AE017282) was cultivated in nitrate mineral salts medium with a head-space of 75% air, 23.75% CH₄ and 1.25% CO₂. Bacteria were grown in 350 ml flasks at 45°C and 200 rpm.

Lactobacillus rhamnosus GG was cultivated in MRS medium (Oxoid) at 37° without agitation. *Escherichia coli* Nissle 1917 (Mutafloor, DSM 6601, serotype O6:K5:H1), was provided by Ardeypharm GmbH, Herdecke, Germany and grown in Luria-Bertani Broth (Oxoid, UK) at 37 °C, 200 rpm. All bacteria were UV inactivated for 60 minutes prior to co-cultivation with MoDCs.

Cells and culture conditions

Buffy coats from healthy volunteers were obtained from Ostfold Hospital Trust, Kalnes, Norway, in accordance with institutional ethical guidelines and with approval from the Regional Committee of Medical and Health Research Ethics. All subjects gave written informed consent in accordance with the Declaration of Helsinki. Peripheral blood mononuclear cells (PBMCs) were isolated by density gradient centrifugation on a Lymphoprep gradient (Fresenius Kabi). CD14⁺ cells were isolated from PBMC by positive selection using human CD14 MicroBeads (Miltenyi Biotec). Immature monocyte-derived dendritic cells (MoDCs) were prepared from CD14⁺ cells by cultivation for 6 days in RPMI 1640 medium supplemented with 10% heat inactivated fetal calf serum, 25 μ g/ml gentamicin sulfate (Lonza), 1 mM sodium pyruvate and 100 μ M non-essential amino acids (both from PAA Laboratories), 25 ng/ml interleukin 4 and 50 ng/ml granulocyte macrophage colony stimulating factor (both from ImmunoTools).

Figure R1: McB lysate lowers plasma TG (A) and total cholesterol (B) in WD-fed mice (12+6 weeks protocol at 22°C).

Point-by-Point response MS_NCOMMS-19-37264

A Final Body Weight

B "Weight matched" Final Body Weight

"Weight matched"

C

Baseline glucose tolerance

D

Baseline insulin

E

GTT Week 12+5

F

Insulin Week 12+5

G

H

Figure R2: A-B: Final bodyweight of A) all and B) a subset of mice included in the 12+6 weeks experiment stratified into ‘weight-matched’ groups. C-H: Glucoregulatory capacity in ‘weight-matched’ mice. C-D: Oral glucose tolerance test (OGTT) and insulin secretion during OGTT, respectively, before intervention; i.e. at baseline, week 11, when all WD-fed mice were fed WD_{REF}, and so differences between groups are based on mouse to mouse variation prior intervention. E-F: As in C-D but after 5 weeks of intervention. G-H: Paired analysis of WD_{REF}- and WD_{M_{CB}}-fed mice before and after their respective intervention. Mice fed WD_{REF} aggravates hyperinsulinemia during OGTT as a function of time. WD_{M_{CB}}-fed mice are protected against this time-dependent trajectory.

References:

Baffy, G. (2009) ‘Kupffer cells in non-alcoholic fatty liver disease: The emerging view’, *Journal of Hepatology*, 51(1), pp. 212–223. doi: 10.1016/j.jhep.2009.03.008.

Campbell, C. *et al.* (2020) ‘Bacterial metabolism of bile acids promotes generation of peripheral regulatory T cells’, *Nature*. Springer US, 581(7809), pp. 475–479. doi: 10.1038/s41586-020-2193-0.

Dal-Secco, D. *et al.* (2015) ‘A dynamic spectrum of monocytes arising from the in situ reprogramming of CCR2+ monocytes at a site of sterile injury.’, *The Journal of experimental medicine*, 212(4), pp. 447–56. doi: 10.1084/jem.20141539.

Furusawa, Y. *et al.* (2013) ‘Commensal microbe-derived butyrate induces the differentiation of colonic regulatory T cells’, *Nature*. Nature Publishing Group, 504(7480), pp. 446–450. doi: 10.1038/nature12721.

Indrelid, S. *et al.* (2017) ‘The soil bacterium *Methylococcus capsulatus* bath interacts with human dendritic cells to modulate immune function’, *Frontiers in Microbiology*, 8(FEB), pp. 1–13. doi: 10.3389/fmicb.2017.00320.

Ju, C. and Tacke, F. (2016) ‘Hepatic macrophages in homeostasis and liver diseases: From pathogenesis to novel therapeutic strategies’, *Cellular and Molecular Immunology*. Nature Publishing Group, 13(3), pp. 316–327. doi: 10.1038/cmi.2015.104.

Kubach, J. *et al.* (2007) ‘Human CD4+CD25+regulatory T cells: Proteome analysis identifies galectin-10 as a novel marker essential for their anergy and suppressive function’, *Blood*, 110(5), pp. 1550–1558. doi: 10.1182/blood-2007-01-069229.

Li, F. *et al.* (2017) ‘The microbiota maintain homeostasis of liver-resident $\gamma\delta$ T-17 cells in a lipid antigen/CD1d-dependent manner’, *Nature Communications*. Nature Publishing Group, 7. doi: 10.1038/ncomms13839.

Odegaard, J. I. *et al.* (2008) ‘Alternative M2 Activation of Kupffer Cells by PPAR δ Ameliorates Obesity-Induced Insulin Resistance’, *Cell Metabolism*, 7(6), pp. 496–507. doi: 10.1016/j.cmet.2008.04.003.

Ramachandran, P. *et al.* (2012) ‘Differential Ly-6C expression identifies the recruited macrophage phenotype, which orchestrates the regression of murine liver fibrosis.’, *Proceedings of the National*

Point-by-Point response MS_NCOMMS-19-37264

Academy of Sciences of the United States of America, 109(46), pp. E3186-95. doi: 10.1073/pnas.1119964109.

Smith, P. M. *et al.* (2013) 'The microbial metabolites, short-chain fatty acids, regulate colonic Treg cell homeostasis.', *Science (New York, N.Y.)*, 341(6145), pp. 569–73. doi: 10.1126/science.1241165.

Song, X. *et al.* (2020) 'Microbial bile acid metabolites modulate gut ROR γ ⁺ regulatory T cell homeostasis', *Nature*. Springer US, 577(7790), pp. 410–415. doi: 10.1038/s41586-019-1865-0.

REVIEWERS' COMMENTS

Reviewer #1 (Remarks to the Author):

Authors have made great effort to follow comments/critiques and have performed the majority of this referee requests and clarifications. Overall the paper has been significantly improved from the original submission.

Reviewer #2 (Remarks to the Author):

All of my comments from the previous review have been satisfactorily addressed.